# Graph Fourier Neural Kernels (G-FuNK): Learning Solutions of Nonlinear Diffusive Parametric PDEs on Multiple Domains

## Abstract

Understanding and predicting the time-dependent dynamics of complex systems governed by non-linear partial differential equations (PDEs), with varying parameters and domains, is a difficult problem that is motivated by applications in many fields. We introduce a novel family of neural operators based on a **G**raph **Fo**urier **N**eural **K**ernel (G-FuNK), for learning solution generators of nonlinear PDEs with varying coefficients, across multiple domains, for which the highest-order term in the PDE is diffusive. G-FuNKs are constructed by combining components that are parameter- and domain-adapted, with others that are not. The latter components are learned from training data, using a variation of Fourier Neural Operators, and are transferred directly across parameters and domains. The former, parameter- and domain-adapted components are constructed as soon as a parameter and a domain on which the PDE needs to be solved are given. They are obtained by constructing a weighted graph on the (discretized) domain, with weights chosen so that the Laplacian on that weighted graph approximates the highest order, diffusive term in the generator of the PDE, which is parameter- and domain-specific, and satisfies the boundary conditions. This approach proves to be a natural way to embed geometric and directionally-dependent information about the domains, allowing for improved generalization to new test domains without need for retraining. Finally, we equip G-FuNK with an integrated ordinary differential equation (ODE) solver to enable the temporal evolution of the system's state. Our experiments demonstrate G-FuNK's ability to accurately approximate heat, reaction diffusion, and cardiac electrophysiology equations on multiple geometries and varying anisotropic diffusivity fields. We achieve low relative errors on unseen domains and fiber fields, significantly speeding up prediction capabilities compared to traditional finite-element solvers.

## 1 Introduction

**Neural Operators for PDEs**   In scientific machine learning, data-driven deep learning methods aim at predicting the solutions of partial differential equations (PDEs), avoiding the computationally expensive numerical integration methods needed for large-scale simulations. This is especially beneficial for applications like domain optimization and precision medicine, where multiple PDEs need to be solved for varying parameters or domains, requiring significant computational resources. Neural operators learn mappings between high-dimensional function spaces, allowing them to generalize across a family of PDEs without retraining for varying parameters or conditions Kovachki et al. (2023). Mathematically, a neural operator $\mathcal{N}_\theta$ is defined as a mapping from an input domain function space $\mathcal{A}(\Omega_\alpha; \mathbb{R}^{d_a})$ to an output target function space $\mathcal{U}(\Omega_\alpha; \mathbb{R}^{d_u})$, represented as

$$\mathcal{N}_\theta : \mathcal{A}(\Omega_\alpha; \mathbb{R}^{d_a}) \to \mathcal{U}(\Omega_\alpha; \mathbb{R}^{d_u}), \tag{1}$$

where $\theta \in \Theta$ denotes the neural operator's parameters, and $\Omega_\alpha \subset \mathbb{R}^d$ (or a $d$-dimensional manifold) represents the spatial domain on which the functions are defined and $\alpha \in \mathscr{A}$ denotes the shape of the domain. The dimensions $d_a$ and $d_u$ denote the respective sizes of the input and output function spaces, often subspaces of Sobolev spaces. Neural operators can be applied to various problems such as those described by continuous functions $C(\Omega; \mathbb{R}^{d_a})$ or Sobolev spaces $H^s(\Omega; \mathbb{R}^{d_a})$ for some $s \geq 0$. The neural operator $\mathcal{N}_\theta$ is an approximation of a true target operator $\mathcal{N}$ (e.g., the solution

operator of a PDE) obtained by training on input-output function pairs $\{a_i, u_i\}_{i=1}^m$, where $a_i \in \mathcal{A}$ and $u_i = \mathcal{N}(a_i) \in \mathcal{U}$. These pairs could be simulation data representing a known, high-fidelity numerical approximation of the PDE. In cardiac electrophysiology, for example, $\mathcal{A}$ might represent the space of initial electrical activation patterns across cardiac tissue, whereas $\mathcal{U}$ could correspond to the resultant electrical potential fields over time.

**Problem Setup** In this context, we focus on the family of second-order nonlinear differential operators, governing the evolution of a function $u : (\mathbf{x}, t) \to \mathbb{R}$, $\mathbf{x} \in \Omega_\alpha$, $t \geq 0$, in the form,

$$
\begin{cases}
\partial_t u(\mathbf{x}, t) = \nabla_{\mathbf{x}} \cdot (\mathbf{K}(\mathbf{x}) \nabla_{\mathbf{x}} u(\mathbf{x}, t)) + \mathcal{S}(u(\mathbf{x}, t), \mathbf{x}, \nabla_{\mathbf{x}} u(\mathbf{x}, t)) = \mathcal{N}(u(\mathbf{x}, t), \mathbf{K}(\mathbf{x}), \mathbf{x}, t), \\
\partial_{\mathbf{n}(\mathbf{x})} u(\mathbf{x}) = 0 , \quad \forall \mathbf{x} \in \partial \Omega_\alpha
\end{cases}
$$
(2)

where $\mathbf{K}$ is a diffusion tensor field on the domain satisfying uniform ellipticity, $\mathcal{N} : \mathbb{R}^{3d} \times \mathbb{R}^{d_{\mathbf{K}}} \to \mathbb{R}$ is a vector function that can encompass source terms, sinks, or other linear and nonlinear interactions within the term $\mathcal{S}$, in this work we use no-flux or no-diffusive-flux Neumann boundary conditions, where $\mathbf{n}(\mathbf{x})$ is the normal to $\partial \Omega_\alpha$ at $\mathbf{x}$, and $\partial \Omega_\alpha$ is the boundary of the spatial domain $\Omega_\alpha$ (in $\mathbb{R}^d$ or a $d$-dimensional manifold $\mathcal{M}^d$) within a family of diffeomorphic domains $\{\Omega_\alpha\}_{\alpha \in \mathscr{A}}$. The dependence of the solution to (2) on the domain $\Omega_\alpha$ is often complex. Note from 2 that we are focusing here on the semilinear case, i.e. $\mathcal{N}$ is linear in its highest-order term (the diffusive component), and in general nonlinear in the lower-order terms.

The training data consists of trajectories $\{u^{(m)}(x_i, t_\ell)\}_{m=1, i=1, \ell=1}^{M, n_{\alpha^{(m)}}, L}$, where $x_i$ is sampled on a graph $\mathcal{G}_{\alpha^{(m)}}$ (a mesh discretizing $\Omega_{\alpha^{(m)}}$), and time points $0 = t_0 < t_1 < \cdots < t_L = T$ are sampled at a fine timescale, together with the parameter $\mathbf{K}^{(m)}$ and the graph $\mathcal{G}_{\alpha^{(m)}}$. In particular, note that the observation at $t = 0$ says that we are given the initial condition, and that different trajectories in the training set may correspond to different (given) values of both the parameter $\mathbf{K}$ and the domain $\mathcal{G}_\alpha$.

The objective (in contrast to equation (1)) is to learn from the training data a *neural* operator $\mathcal{N}_\theta$, with $\theta$ denoting the neural network parameters, that approximates the *generator* of the solutions of (2) by learning a mapping which takes any given state $u(x, t_\ell)$, parameters $\mathbf{K}$ and domain $\mathcal{G}_\alpha$, and outputs an approximation of the left-hand side of (2). Solutions of the PDE may be approximated by integrating

$$
\partial_t u(x, t) \approx \mathcal{N}_\theta(u(x, t); \mathcal{G}_\alpha, \mathbf{K}) ,
$$
(3)

with $x \in \mathcal{G}_\alpha$, on the whole interval $[0, T]$, given any initial condition $u(\cdot, 0)$ even with changes in parameters and spatial domain. This solver has the chance of being more efficient than one for equation 2, for example by employing larger time steps, by efficiently incorporating the shape of $\Omega_\alpha$ and the boundary conditions, or by reducing the dimensionality of the problem when $\mathbf{x}$ is high-dimensional.

**Reduced Models and Homogenization** In fact, we are very much interested in situations where the "true" system is driven by equations more general than equation 2, with many more variables, possibly evolving at multiple time scales; this will be the case in examples 2 and 3, where besides the spatial variables and the unknown action potential $u$, there are many other quantities $\mathbf{v}$ evolving in space and time (e.g. modeling ionic concentrations that affect the evolution of $u$, and these are driven by a large system of ODEs). In these situations, given observations from such systems, our estimated equation 3 can be viewed as performing dimension reduction on the variables (e.g. by keeping only the spatial variables) and homogenization, obtaining an effective equation for $u$, without tracking the evolution of the other quantities $\mathbf{v}$.

**Related Works** Several studies have demonstrated the ability of different types of neural networks to approximate nonlinear operators with strong theoretical guarantees (e.g., PDE solution operator) Chen & Chen (1995); Lu et al. (2021); Kovachki et al. (2021). Fourier Neural Operators (FNOs) and Deep Operator Networks (DeepONets) and their variants are among the most popular computational frameworks under the umbrella of neural operators Lu et al. (2021); Li et al. (2020a). Vanilla DeepONets, however, are restricted to a fixed grid and resolution, and thus do not generalize well on unseen domains. Yin et al. (2024) propose a work-around spatial transformations of multiple domains to a universal domain where DeepONets can be used to learn a latent operator, the evaluation of which can be mapped back to the target domains via the inverse for evaluation. FNOs, while robust

to variations in grid resolution, require regularly spaced square grids to perform FFT Li et al. (2020a). As an improvement, Li et al. (2023) propose a Geometry-Aware Fourier Neural Operator (Geo-FNO), which learns an additional deformation step to transform irregular grids, achieving greater accuracy than interpolating the solution on a uniform grid. However, they explore only a relatively small subset of possible transformations and do not investigate generalization performance on differently shaped grids. For large-scale 3D PDEs, Li et al. (2024) also propose a more general way of adapting to arbitrary domains and irregular grids with Geometry Informed Neural Operators (GINO) which incorporates information about the geometry into the learning process, however, they mention that the trained model is limited to certain family of shapes and to generalize well, abundant, geometrically varying training samples are necessary, a luxury that is not available in real world settings such as computational medicine. Several works have also shown promise with regards to predicting full solution trajectories over a time interval, but typically do not carry the same generalizability over multiple domains as the aforementioned works Chen et al. (2023); Zhang et al. (2024); He et al. (2024); Regazzoni et al. (2024).

Graph Neural Networks (GNNs) are a versatile class of neural network architectures designed to work directly with graph-structured data. These networks can make predictions at the level of nodes, edges, or entire graphs, typically utilizing a method known as message passing Gilmer et al. (2020). In this method, neighboring nodes exchange messages, and the aggregated information from these messages is used to update the states of subsequent nodes. GNNs are particularly well-suited for irregular grid data and can be effectively applied to training data generated by finite-element method simulations (considered ground truth). Li et al. (2020b) proposed a message-passing operator (using edge-conditioned graph convolutions Simonovsky & Komodakis (2017)) to learn solution operators for PDEs. Iakolev et al. also utilized a message-passing scheme to learn isotropic PDEs Iakovlev et al. (2020). Other studies have considered learning time-dependent PDEs using graph-based approaches Li et al. (2020c); Behmanesh et al. (2023); Pilva & Zareei (2022), but none of these works combine learning with both varying domains and varying parameters in the PDE. Furthermore, the message-passing frameworks they employed in these works involve edge aggregation schemes that limit the learning of direction-dependent information, which is crucial in the case, for example, of anisotropic diffusions. Finally, several studies have shown that spectral GNNs possess superior expressiveness and interpretability, capturing global information more effectively than spatial GNNs Bo et al. (2023); Defferrard et al. (2016); Wang & Zhang (2022); Yang et al. (2022).

**Contributions**    We propose a new family of neural operators consisting of novel **G**raph **Fo**urier Neural **K**ernel (G-FuNK) layers for learning generators of solutions to time-dependent PDEs with varying coefficients across multiple diffeomorphic domains as in equation (2). This method combines ideas from both GNNs and FNOs: it leverages the spectral domain of graphs, unlike GNNs that rely on localized message passing, and is suitable on general graphs rather than relying on grids, as FNOs. G-FuNK employs the Graph Fourier Transform (GFT) to convert functions on graphs into functions on the graph spectral domain. This facilitates global information integration across the entire graph, allowing G-FuNK to capture both local and global dependencies effectively, as FNOs do in the classical Euclidean grid setting.

G-FuNK integrates components that are both parameter- and domain-adapted with others that are not. The non-adapted components are learned from training data, and can be directly transferred across different parameters and domains, independent of the mesh resolution or the time step size. The adapted components are constructed for specific parameters and domains by forming a parameter-dependent, tailored weighted graph on the discretized domain, such that the corresponding graph Laplacian approximates the highest-order diffusive term in the PDE with those specific values of the parameter, on that domain.

We remark that here we do wish to predict *entire trajectories* of temporal dynamics on $[0, T]$, which is difficult as inaccuracies can lead to dramatic accumulation of errors, depending on the PDE. Many existing methods Li et al. (2020a;b;c) are either restricted by design on learning the map from initial conditions (or several observation points) to a solution at a fixed time $t$, or while they could in principle generate solutions at all times, experiments in this setting are not reported in the corresponding papers. With this goal in mind, our framework is equipped with an integrated ODE solver to enable generating the temporal evolution of the dynamics of the system. This allows for efficient, mesh-independent prediction of the system's solutions from new initial conditions, with new values of the parameter $\mathbf{K}$, on a new domain $\Omega_\alpha$. To our knowledge, there is no current neural

operator framework which naturally handles directionally dependent information of anisotropic domains while predicting time-evolving trajectories of the solution operator on multiple geometries.

We demonstrate G-FuNK's capabilities by first considering the basic example of the anisotropic heat equation on the square, where the tensor field $\mathbf{K}(\mathbf{x})$ reflects the directional dependence of diffusivity. We then move to nonlinear, more complex models of reaction-diffusion equations for cardiac electrophysiology (EP), first on families of rectangular domains, then on 3D geometries representing the outer surfaces of real human left atrial (LA) chambers from computed tomography (CT) data. The PDE operators on the patient-specific LA chambers involve effects from high-dimensional coupled ODEs. Our method accurately captures the complex dynamics of the transmembrane potential on varying geometries and anisotropic diffusivity (fiber) fields. Our experiments show that G-FuNK can significantly speed up patient-specific cardiac electrophysiology simulations, which are used for making real-time quantitatively informed clinical decisions Boyle et al. (2019); Trayanova et al. (2024), achieving low relative errors on unseen domains.

## 2 G-FuNK: Graph Fourier Neural Kernel

### 2.1 Graphs, Laplacians, and the Graph Fourier Transform

Let $\mathcal{G}_\alpha(\mathcal{V}, \mathcal{E})$ be an undirected graph where $\mathcal{V} = \{x_i\}_{i=1}^{n_\alpha}$ is the set of $n_\alpha$ nodes (or vertices) constructed from a finite element mesh discretization or down-sampled point cloud representation of $\Omega_\alpha$ and $\mathcal{E} \subseteq \{(x_i, x_j) | x_i, x_j \in \mathcal{V}\}$ is the set of edges that connects pairs of nodes which are defined as the k-nearest neighbors of each $x_i$. Let $W \in \mathbb{R}^{n_\alpha \times n_\alpha}$ be the weighted adjacency matrix, with $W_{ij} = W_{ji} \geq 0$, with strict inequality if and only if $(x_i, x_j) \in \mathcal{E}$, and deg is the $n_\alpha \times n_\alpha$ diagonal degree matrix of the graph, defined by $\deg_{ii} := \sum_j W_{ij}$. The undirected graph Laplacian is a positive semi-definite matrix with a spectral decomposition given by

$$\mathscr{L}_{\mathcal{G}_\alpha} := \deg - W = \Psi \Lambda \Psi^T, \tag{4}$$

where $\Psi = [\psi_1, \ldots, \psi_n] \in \mathbb{R}^{n_\alpha \times n_\alpha}$ is the matrix of orthonormal eigenvectors of $\mathscr{L}_{\mathcal{G}_\alpha}$, and $\Lambda = \text{diag}(\lambda_1, \lambda_2, \ldots, \lambda_{n_\alpha})$ is the diagonal matrix of the corresponding non-negative eigenvalues. Note that $\{\psi_i\}_{i=1}^{n_\alpha}$ is an orthonormal basis for functions on the graph, since $\mathscr{L}_{\mathcal{G}_\alpha}$ is symmetric. When the graph is constructed on points randomly sampled on a domain or a manifold, this discrete Laplacian is an approximation (in a suitable sense) to a continuous Laplacian in the limit as the number of samples goes to infinity, and its eigenvectors are approximations to the eigenfunctions of the continuous Laplacian, which are the natural generalization of Fourier modes from a torus or rectangular box to general domains and manifolds. Furthermore, the geometry of the domain (graph, in the discrete case, Riemannian manifold in the continuous case) is completely determined by the Laplacian, and therefore by its eigenvalues and eigenvectors.

We define the Graph Fourier Transform (GFT) as follows: for a function $a \in \mathbb{R}^{n_\alpha}$ on the vertices of $\mathcal{G}_\alpha$, the Graph Fourier Transform (GFT), denoted by $\mathcal{F}$, rewrites $a$ in the basis of eigenvectors of the graph Laplacian $\mathscr{L}_{\mathcal{G}_\alpha}$, yielding a function $\hat{a}$ of the eigenvalues $\lambda_1, \ldots, \lambda_{n_\alpha}$, defined as

$$\hat{a} := \mathcal{F}a := (\langle \psi_i, a \rangle)_{i=1,\ldots,n_\alpha} = \Psi^T a, \tag{5}$$

In the reverse process, the inverse Graph Fourier Transform (IGFT), represented by $\mathcal{F}^{-1}$, reconstructs the function $a$ from its spectral representation $\hat{a}$, through the equation

$$a = \mathcal{F}^{-1}\hat{a} := \sum_{i=1}^{n_\alpha} \hat{a}(\lambda_i)\psi_i = \Psi \hat{a}. \tag{6}$$

It should be noted that computing the $k_{\max}$ lowest eigenvalues for an undirected graph is generally inexpensive with a complexity of $\mathcal{O}(k_{\max}^2 n_\alpha j)$ for a graph with $n_\alpha$ vertices, $j$ neighbors per vertex. Eigenpairs are of course computed once per domain/diffusivity parameter, rather than being recomputed at every instance during network training.

Graph diffusion is analogous to the heat equation and describes the diffusive movement of a function across a graph. The solution to the graph diffusion equation can be expressed in the basis of the

spectral decomposition of the graph Laplacian $\mathscr{L}_{\mathcal{G}_\alpha}$ as

$$\frac{\partial a(t)}{\partial t} = c\mathscr{L}_{\mathcal{G}_\alpha}a(t), \quad a(t) = \sum_{i=1}^{n_\alpha} \mathcal{F}(a(0))_i e^{-c\lambda_i t}\psi_i \tag{7}$$

where $c$ is the diffusivity constant, and $a(t) \in \mathbb{R}^{n_\alpha}$ is the function at time $t$, and the initial condition is the function $a(0) = (a_i(0))_{i=1}^{n_\alpha}$. The solution can be truncated to the $k_{\max}$ lowest modes, where $k_{\max} \leq n_\alpha$, for an approximated solution. It should be noted that the *form* of Eq. (7) is not dependent upon a spatial coordinate but only on the spectral decomposition of $\mathscr{L}_{\mathcal{G}_\alpha}$.

By establishing these preliminaries, we set the stage for the development of G-FuNK, which leverages the spectral properties (precomputed lowest eigenpairs) of the graph Laplacian to efficiently learn and generalize dynamic processes across different domains.

## 2.2 THE G-FUNK LAYERS AND NETWORK

In this section, we describe the neural operator framework, a variation of the FNO of Li et al. (2020a), equipped with our novel **G**raph **Fo**urier **N**eural **K**ernel (G-FuNK) layers to learn solution generators to time-dependent PDEs across multiple domains and multiple anisotropic diffusion tensor fields.

Recall that the domain – or, rather, a discretization thereof – is given, and so is the diffusion field **K**. We exploit the knowledge of the former to construct a graph $\mathcal{G}_\alpha$ which discretizes $\Omega_\alpha$ into $n_\alpha$ nodes $\mathcal{V} = \{x_i\}_{i=1}^{n_\alpha}$, and knowledge of the latter to weight the edges in order to approximate the second-order term $\nabla \cdot (\mathbf{K}(\mathbf{x})\nabla)$ in the PDE. By choosing, on each edge $\mathcal{E}_{ij}$,

$$w_{ij}^{-1} := \frac{1}{2}(x_j - x_i)^T \cdot (\mathbf{K}(x_i)^{-1} + \mathbf{K}(x_j)^{-1}) \cdot (x_j - x_i), \tag{8}$$

we weight the edges at $x_i$ proportionally to the direction of the diffusion coefficient $\mathbf{K}(x_i)$ at $x_i$, and to the squared inverse of the distance. The average of $\mathbf{K}(x_i)^{-1}$ and $\mathbf{K}(x_j)^{-1}$ is used to obtain a symmetric graph.

The neural operator $\mathcal{N}_\theta$, equipped with several G-FuNK layers, approximates the solution generator of the PDE by applying a series of transformations, which can be formally expressed as the composition of different functions representing the network layers. The network begins with the following function transformation:

$$k_0(x) = \mathcal{P}(a(x); \theta_P), \tag{9}$$

where $\mathcal{P} : \mathbb{R}^{n_\alpha \times d_a} \to \mathbb{R}^{n_\alpha \times d_p}$ is the initial lifting operation to a higher dimensional feature space.

For each G-FuNK layer $n = 1, \ldots, N$, we have, following the structure diagram from left to right:

$$\begin{aligned}
\hat{k}_n(\lambda) &= \mathcal{F}(k_n(x); \Psi^T)(\lambda), \\
\hat{\ell}_n(\lambda) &= \mathcal{L}_n(\hat{k}_n(\lambda); B)(\lambda), \\
\hat{r}_n(\lambda) &= \mathcal{R}_n(\hat{\ell}_n(\lambda); \theta_{\mathcal{R}_n})(\lambda), \\
\hat{f}_n(x) &= \mathcal{F}^{-1}(\hat{r}_n(\lambda); \Psi)(x), \\
\omega_n(x) &= \mathcal{W}_n(k_n(x); \theta_{\mathcal{W}_n})(x), \\
z_n(x) &= \sigma(\hat{f}_n(x) + \omega_n(x))(x) = k_{n+1}(x),
\end{aligned} \tag{10}$$

where $\mathcal{F} : \mathbb{R}^{n_\alpha \times d_p} \to \mathbb{R}^{k_{\max} \times d_p}$ represents the Graph Fourier Transform that projects the function into the $k_{\max}$ lowest frequencies in the spectral domain, $\mathcal{L}_n : \mathbb{R}^{k_{\max} \times d_p} \to \mathbb{R}^{k_{\max} \times d_p \times p}$ for $n = 1, \ldots, N$ is a linear transformation of $\hat{k}_n$ together with a set of the eigenvalues raised to $p$ powers (i.e. $B$ is a matrix $\mathbb{R}^{k_{\max} \times p}$ with columns of $(\lambda_1^0, \ldots, \lambda_{k_{\max}}^0), (\lambda_1^1, \ldots, \lambda_{k_{\max}}^1), \ldots, (\lambda_1^{p-1}, \ldots, \lambda_{k_{\max}}^{p-1}))$, $\mathcal{R}_n : \mathbb{R}^{k_{\max} \times d_p \times p} \to \mathbb{R}^{k_{\max} \times d_{p'}}$ for $n = 1, \ldots, N$ are the parameterized linear transformations in the spectral domain corresponding to the $n$-th G-FuNK layer which can be diagional, tri-diagonal, full matrices, etc., $\mathcal{F}^{-1} : \mathbb{R}^{k_{\max} \times d_{p'}} \to \mathbb{R}^{n_\alpha \times d_{p'}}$ is the inverse Graph Fourier Transform that maps the transformed spectral function back to the graph domain. $\mathcal{W}_n : \mathbb{R}^{n_\alpha \times d_p} \to \mathbb{R}^{n_\alpha \times d_{p'}}$ for $n = 1, \ldots, N$ are additional linear mappings applied after transforming back to the graph domain in each G-FuNK layer, $\sigma : \mathbb{R}^{n_\alpha \times d_{p'}} \to \mathbb{R}^{n_\alpha \times d_{p'}}$ an activation function applied in each G-FuNK layer.

After processing through all $N$ G-FuNK layers, the final output is obtained by the projection layer:

$$y(x) = \mathcal{Q}(z_N(x); \theta_\mathcal{Q}) \tag{11}$$

where $\mathcal{Q} : \mathbb{R}^{n_\alpha \times d_{p'}} \to \mathbb{R}^{n_\alpha \times d_u}$ projects from the last layer's output onto the target space.

In the G-FuNK framework, the eigenvectors of the Laplacian perform the same action as the Fourier transform in the FNO framework, with the important difference that these eigenvectors are adapted to the diffusion coefficient, capturing the parametric dependence of the highest order differential term in the PDE. We suggest using the true eigenfuctions of the shape which embed global structural and physical properties as corroborated by numerical and statistical methods which represent PDE solutions in terms of known basis functions that contain information about the solution structure Bhattacharya et al. (2021); Nagy (1979); Almroth et al. (1978). Furthermore, in FNOs, the eigenvalues are not needed as there is no domain change; here by incorporating the eigenvalues we capture both the dependency on $\mathbf{K}$ and on the domain $G_\alpha$. Finally, we comment that $\mathcal{L}_n$ and $\mathcal{R}_n$ allow G-FuNK to approximate spectral multipliers, while $\mathcal{W}_n$ allows for the approximation of spatial pointwise multipliers. At high level, this is particularly well-suited for reaction-diffusion equations, where the diffusion component is expected to be easy to learn through spectral multiplier, and the nonlinear reaction terms act pointwise and can be expected to be captured by the spatial multipliers. Of course such PDEs have complex interactions between the spatial and spectral domain, so this intuition is very much qualitative.

All together, the neural operator with $N$ G-FuNK layers can be expressed as the following composition of the afforementioned transformations to approximate $\partial_t u(\mathbf{x}, t)$:

$$\mathcal{N}_\theta(a(x)) := \mathcal{Q} \circ \underbrace{\sigma(\mathcal{W}_N + \mathcal{F}^{-1} \circ \mathcal{R}_N \circ \mathcal{L}_N \circ \mathcal{F})}_{\text{G-FuNK Layer } N} \circ \cdots \circ \underbrace{\sigma(\mathcal{W}_1 + \mathcal{F}^{-1} \circ \mathcal{R}_1 \circ \mathcal{L}_1 \circ \mathcal{F})}_{\text{G-FuNK Layer } 1} \circ \mathcal{P}(a(x); \theta),$$

$$\tag{12}$$

We denote $\theta = \{\theta_{\mathcal{P}_n}, \theta_{\mathcal{R}_n}, \theta_{\mathcal{W}_n}, \theta_{\mathcal{Q}_n}\}_{n=1}^N$ as a collection of all learnable parameters in the network that are optimized during the training phase to minimize the discrepancy between the neural operator's output and the known numerically computed trajectories. Figure 1 illustrates a summary of the network architecture.

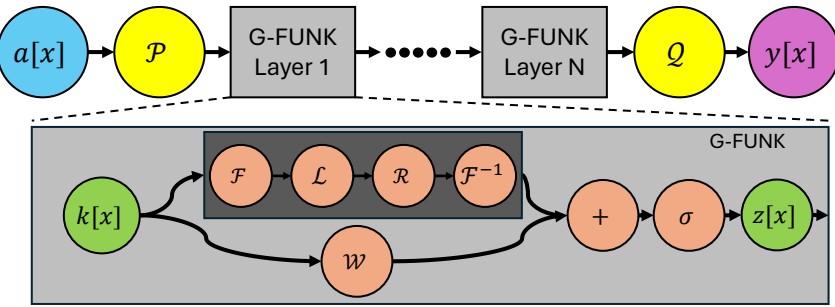

Figure 1: **Top:** Network structure of the Neural Operator. The input is passed as $a$. (1) It is lifted to a higher dimensional space via a lifting operator $\mathcal{P}$. (2) Multiple G-FuNK layers are then applied. (3) The network projects to the output dimensional space via $\mathcal{Q}$. **Bottom:** Single G-FuNK layer structure. Starting from input $k$, (1) $\mathcal{F}$ denotes the GFT of the input which is then reduced to the $k_{\max}$-lowest modes. (2) The function is expanded onto a set of the eigenvalues raised to different powers in $\mathcal{L}$ (3) A linear transform is applied with learnable parameters in $\mathcal{R}$. (4) $\mathcal{F}^{-1}$ represents the inverse GFT, mapping back to the original domain. (5) The input function $k$ undergoes a linear mapping in $\mathcal{W}$. (6) The outputs of the top and bottom branches are combined and then passed through an activation function $\sigma$. The output of the G-FuNK layer is $z$ and is operated on by the next G-FuNK layer.

Given discrete observations $\{u(x_i, t_\ell)\}_{\ell=1}^L$ of the state variable at a location $x_i \in \Omega_\alpha$ at different time steps $0 = t_1 < \cdots < t_L$, we can integrate the approximation over time using an ODE solver, $\mathcal{I}$:

$$u(x_i, t_\ell + \Delta t) = \mathcal{I}(\mathcal{N}_\theta(u(x_i, t_\ell, \Psi, \Lambda)), \tag{13}$$

where $\Delta t$ is the time step size, and $\mathcal{I}$ is a neural ODE numerical algorithm (we use the one proposed by Chen et al. (2018)). This integrator uses the adjoint method during the gradient computation in backpropagation (since the loss function involves predicting the solution at multiple future time points), which makes training computationally demanding.

**Mesh-Independence of G-FuNK** Although the proposed framework involves mesh-dependent steps, such as forming a weighted graph on the discretized domain, the method maintains a high degree of mesh-independence. More precisely, as the number of points increases, the Laplacian we construct approaches its limit, allowing our framework to effectively handle finer discretizations. Moreover, there are established methods for interpolating eigenfunctions to an underlying continuous domain Coifman et al. (2005), which further supports the transition from discrete to continuous representations. This characteristic is crucial, as it sheds light on the ability of our method to transition from discrete approximations to continuous domains as the mesh size approaches zero. Our framework's capability to interpolate and generalize across varying mesh sizes ensures that it remains robust and accurate in capturing the underlying dynamics of PDEs in the limit of fine discretizations.

## 3 NUMERICAL EXPERIMENTS

In this section, we present several numerical experiments demonstrating the ability of the G-FuNK operator learning framework to accurately predict the solution generator for PDEs of the form in (2), towards the goal of accelerating precision medicine in cardiac electrophysiology. Results for all examples and comparisons to baseline models are summarized in Table 1.

**Heat Equation**. The introduction of anisotropic diffusion is critical in simulating phenomena such as the propagation of heat in materials with fibrous structures, where the diffusive properties vary along different directions. We consider the classical heat equation with very strong anisotropic diffusion on a 2D unit square domain with no-flux Neumann boundary conditions, with $\mathbf{K}$ as a diffusion tensor representing the anisotropy:

$$\partial_t u = \nabla \cdot (\mathbf{K}(\mathbf{x})\nabla u), \quad \mathbf{x} \in \Omega_\alpha, t > 0, \quad \partial_{\mathbf{n}(x)} u(\mathbf{x}) = 0 \; \forall \, \mathbf{x} \in \partial\Omega_\alpha \tag{14}$$

In our experiments, we consider a diffusion tensor with an anisotropy ratio of 9:1, given by:

$$\mathbf{K}(\mathbf{x}) = \begin{bmatrix} \hat{F}^{(1)}(\mathbf{x}) & \hat{F}^{(2)}(\mathbf{x}) \end{bmatrix} \begin{bmatrix} 9 & 0 \\ 0 & 1 \end{bmatrix} \begin{bmatrix} \hat{F}^{(1)}(\mathbf{x}) & \hat{F}^{(2)}(\mathbf{x}) \end{bmatrix}^T, \tag{15}$$

where $\hat{F}^{(1)}(\mathbf{x})$ and $\hat{F}^{(2)}(\mathbf{x})$ are orthonormal vectors fields that define the longitudinal and transverse directions of diffusion and are described in Appendix section A.1.1, equation (17). The longitudinal direction of the anisotropic diffusion were defined as a superposition of geometric and linear functions, with random parameters sampled independently for each trajectory in both training and test data.

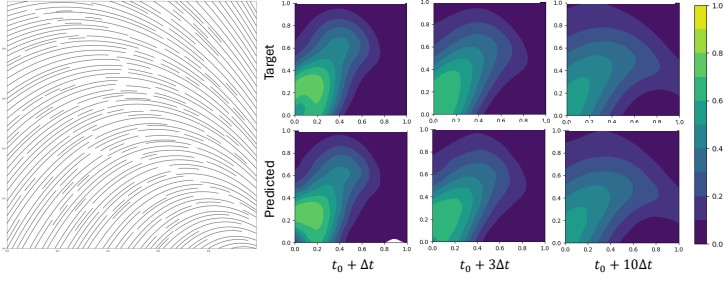

Figure 2: **Heat Equation: G-FuNK vs Target.** A comparison between the target and the G-FuNK predicted heat equation. On the left is a stream plot example of the primary direction of the diffusion, described as $\hat{F}^{(1)}$ in (17), which was unique for the test trajectory shown on the right. The prediction of G-FuNK is on the bottom and the target is above for three different time points after the initial condition at $t_0$.

For this case, the eigenvalues in $B$ of (10) were raised to only the $0^{\text{th}}$ power, and fiber information was not used as an input to the network. Therefore, learning the influence of anisotropic diffusion was done using only the influence of the eigenvectors. For this case, $\theta_R$ was only a diagonal matrix of learnable parameters. Figure 2 contains a depiction of a diffusion field $\hat{F}^{(1)}$. Additional information about the network can be found in A.1.

**2D Nonlinear Reaction Diffusion**   We consider a reaction diffusion system in a 2D domain with no-diffusive-flux Neumann boundary conditions. This system of equations is very stiff and involve multiple non-linear equations to describe the propagation of action potentials in the form of a bioelectrical wave Courtemanche et al. (1998). The solutions of these equations typically have the range between $-85$ and $20$ mV, with the wavefront having a rate of change $> 100 \frac{mV}{ms}$, making them steep and close to discontinuous. For these systems, an external stimulus current is applied to start the wave propagation from a random point within the domain for each trajectory.

In terms of the transmembrane potential $u$, the system is of the form

$$\partial_t u = \nabla \cdot (\mathbf{K}(\mathbf{x})\nabla u) + \sum_s J_s(u, \mathbf{v}), \quad \frac{d\mathbf{v}}{dt} = \Upsilon(u, \mathbf{v}), \quad \mathbf{K}(\mathbf{x})\nabla u \cdot \mathbf{n}(\mathbf{x}) = 0 \, \forall \mathbf{x} \in \partial\Omega_\alpha \,, \quad (16)$$

where $\mathbf{x} \in \Omega_\alpha, t > 0$, $\sum_s J_s$ is the sum of 12 ionic and external stimulus currents described by non-linear functions Courtemanche et al. (1998). In this case, no diffusive flux Neumann boundary conditions were used.

In this example, each trajectory had a unique rectangular domain, $\Omega_{\alpha(m)}$, with edge length drawn independently from 2 uniform distributions on $[15, 30]cm$. The diffusive tensor $\mathbf{K}(\mathbf{x})$ was set at 5 to 1 in the x direction across each domain. Results are shown in Figure 3. Additional information about the network can be found in A.2.

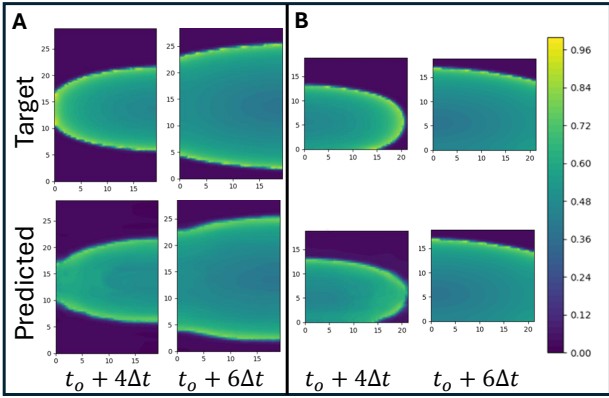

Figure 3: **Reaction Diffusion on Random Rectangle: G-FuNK vs Target. A** and **B** are two comparisons of G-FuNK for 2 different test geometries. Both panels use the colorbar on the right which is min-max scaled from -85 to 20 mV and $\Delta t$ is 10 milliseconds.

**Cardiac Electrophysiology**. In the field of personalized cardiac electrophysiology (EP), the ability to describe the electrical propagation in the Left Atria (LA) is of the utmost importance to provide better healthcare to patients inflicted by electrical abnormalities. Using cardiac computed tomography (CT) scans from 25 atrial fibrillation (AF) patients, triangulated meshes of the LA surface were constructed from image segmentations. The mitral valve and four pulmonary veins were removed via visual inspection resulting in a topology with five holes. The fiber orientation for each mesh was mapped using an atlas-based procedure Roney et al. (2021); Ali et al. (2021); Roney et al. (2019) and produced fibers similar to the example shown in Appendix A.4.2 Figure 9. The process of manually annotating the geometry to define the fiber fields adds inherent noise from one geometry to another.

Finite element simulations were computed using the openCARP software Plank et al. (2021) in which the Courtemanche electrophysiological ionic model Courtemanche et al. (1998) was solved with no-flux Neumann boundary conditions to describe the electrical wave propagation in each domain. The Courtemanche equations follow the same form as (16). The diffusive ratio was set as 5 to 1 in the fiber direction with a base diffusive value of $0.625 \frac{mS}{mm}$. Each simulation incorporated a stimulus current, represented as a delta spike, chosen at a random location within the domain and the initial condition was chosen at 10 ms after this stimulus was applied. For this work, the aim was to start from an initial condition $u_0$ and compute the next time step repeatedly. This was done using 24 domains, G-FuNK was trained to infer the solution from 0ms to 90ms which focused on the steep

wavefront, and an additional different geometry was used as an out-of-training test set as shown in Figure 4. Information about the network can be found in A.3.

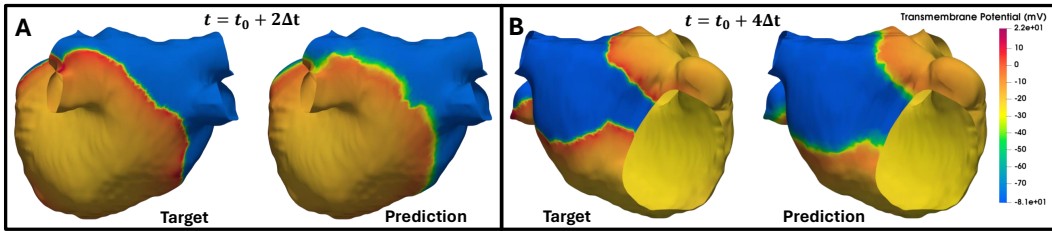

Figure 4: Comparing target and G-FuNK predictions on an out-of-training test geometry. **A** Left atrial posterior wall view at $2\Delta t$ (20 ms) after initial condition. **B** Left atrial anterior wall view at $4\Delta t$ (40 ms) after initial condition.

**Results** The performance of G-FuNK and other models is summarized in Table 1 below.

| Problem | Geometries | **G-FuNK** Parameters Rel $\ell_2$ | FNO Parameters Rel $\ell_2$ | GNN Parameters Rel $\ell_2$ |
|---|---|---|---|---|
| Heat Equation | 2D Unit Square Varying Diffusion Training Trajectories: $M = 500$ (GNN: $M = 1000$) | 197465 0.0357 | FNO 2017157 **0.0134** | *With Edge Weights* 12226 0.1875 *Without Edge Weights* 35026 0.2428 |
| Reaction Diffusion | 2D Random Rectangles No Varying Diffusion Training Trajectories: $M = 1000$ | 134965 0.1189 | Geo-FNO 2515307 **0.1162** | *With Edge Weights* 20751 0.3345 |
| | Test Trajectories Rotated $90°$ | Invariant **0.1189** | Not Invariant 0.5681 | Not Invariant 0.3663 |
| Cardiac EP | Multiple 3D Patient Geometries Varying Diffusion Training Trajectories: $M = 1000$ | 283040 **0.1642** | Not Applicable | *With Edge Weights* 248851 0.4034 |

Table 1: Performance of G-FuNK, FNO (or Geo-FNO) and a GNN equipped with a neural ODE on the different problems presented in 3. The performance is on a test set with new initial conditions and, where applicable, new domains and diffusive fields not seen during training.

The Heat Equation model was tested with varying amounts of trajectories and datapoints per sample. The resulting performances are reported in Table 2 and 3 in Appendix A.1.3.

**Discussion** Altogether, G-FuNK was adaptable to all the examples discussed above with low relative error. The results for the various tasks are listed in Table 1. G-FuNK performed well on all examples with a fairly small amount of learnable parameters compared to FNOs and GeoFNOs Li et al. (2020a; 2024).

Message-passing GNNs aggregate information at the nodes, and as expected are unable to accurately handle directionally dependent information (varying diffusive fields). We notice that when we incorporate edge weights, as defined in equation (8), into the GNN model for the Heat equation example, the results improve and the model requires much fewer parameters. Therefore, for all comparisons following, we incorporated the edge weights as an input into the different networks.

Prior works such as FNO and GeoFNO, with the code provided, do not present results or compatibility for learning full trajectories of time-evolving processes, where errors can accumulate due to spatial misalignment of predictions Li et al. (2020a; 2023). To compare with FNO and GeoFNO, we modified the approaches to be compatible with the same neural ODE solver we use for time-dependent predictions with G-FuNK. For the heat example, it should be noted that G-FuNK was able to learn the

effects of anisotropic diffusion from only the eigenvectors. The other networks were given the primary diffusive vector as inputs at each sample point which has less meaning outside of planar geometries. In the 2D reaction-diffusion problem with random rectangles, both GeoFNO and G-FuNK perform similarly on the test set, as expected. In regular rectangular domains, the graph Fourier and fast Fourier bases converge, implying that FNO and GeoFNO are subsets of G-FuNK equipped neural operators, corroborated by the similar performance. For training in this example, we still considered anisotropy, but with all the fiber fields strictly pointing in the horizontal axis. We show that G-FuNK and GeoFNO perform quite similarly on test domains with the same directionality presented in the fiber fields as the training samples. However, when the test set domains and fiber fields were rotated by 90 degrees, GeoFNO's performance significantly degraded, with a relative $\ell_2$ error of 0.5681, as shown in Table 1 and Figure 7 shown in the Appendix A.2.4. In contrast, G-FuNK, trained using only the initial condition and graph eigenpairs, remained rotation-invariant to the unseen fiber orientation.

For the cardiac EP example, we do not show results with GeoFNO since the complex 3D geometries of the human left atrium with five holes cannot be mapped diffeomorphically to a cube or torus where the Fourier transform is available. For this example, our G-FuNK equipped operator learning framework is able to provide respectable predictions, where a majority of the relative $\ell_2$ error is due a small lag in the wavefront of about 1.62 ms and the wavefront being slightly more diffusive. This shift resulted in a higher $\ell_2$ error but the cross-correlation is $0.941$ implying that the prediction is correctly shaped with a small lag. Increasing the number of geometries in the training set will decrease this error as we only use 24 geometries for training since developing the geometries and simulations can take over 1 day per patient.

Our method predicts entire trajectories in under 1 second, significantly outperforming traditional numerical methods. For example, cardiac EP simulations take on average 13.2 (min: 8.12, max 21.97) minutes on 12 CPU cores for one given set of initial conditions. To make quantitatively informed clinical predictions, one must perform comprehensive parameter sweeps to identify optimal treatment strategies for a given patient, which could take hours to days with the finite element approach. These trajectories are predicted by the G-FuNK equipped operator learning framework in less than 1 second. The inclusion of more geometries in the 3D cardiac EP examples is of course expected to lead to lower error. Additionally, small changes in the eigenvalues across domains can lead to mismatches in the order of the eigenvalues between geometries which could be a source in the reported error: this can be avoided with an eigenvector-matching procedure, which will be the subject of future investigations. At the moment, the predictions of our model are restricted to test data from the same sample distribution as the training data. Out-of-distribution predictions are proviced in Appendix A.2.3. We believe that an eigenvalue matching procedure as mentioned above may improve the ability of G-FuNK's ability as an extrapolator.

## 4   CONCLUSION

Data-driven methods for PDEs are highly dependent on the quality and quantity of the data provided. In this work, we present a new family of neural operators based on our novel Graph Fourier Neural Kernel (G-FuNK) layers which combine GNNs and FNOs. We demonstrate that our framework achieves high accuracy on pedagogical examples of the anisotropic 2D heat and reaction-diffusion equations. Even in the relatively data-starved setting of cardiac EP considered here, the accuracy of G-FuNK is respectable. Altogether, the proposed method leverages the expressiveness and versatility of learning in the graph spectral domain. Our G-FuNK-equipped operator learning framework performs well on unstructured data to efficiently and accurately predict the temporal dynamics of complex systems from a single initial condition with varying anisotropic diffusion tensor fields, highly nonlinear reaction terms, and different domains using real cardiac patient data.

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

# A APPENDIX

## A.1 HEAT EQUATION

### A.1.1 HEAT DATA GENERATION

For this section of work, the heat equation was solved using the `NDSolve` function from the Mathematica software on the 2D unit square where the initial condition, $u_0$, was unique for each trajectory and chosen as a superposition of three binomial distributions where each center, deviation, and covariance were picked from uniform distributions with ranges of $[0, 1]$, $[.1, .2]$ and $[-.1, .1]$, respectively. The primary direction of diffusion was defined as $F^1$ in the equation below.

$$F^{(1)}(x, y) = \begin{bmatrix} \sin(a_1(\frac{x+a_2}{2\pi})) + \cos(a_3(\frac{y+a_4}{2\pi})) + a_5 x + a_6 + a_7 y + a_8 \\ \sin(b_1(\frac{x+b_2}{2\pi})) + \cos(b_3(\frac{y+b_4}{2\pi})) + b_5 x + b_6 + b_7 y + b_8 \end{bmatrix} \tag{17}$$

The parameters $a_1$ to $a_8$ and $b_1$ to $b_8$ were derived from the following uniform distributions and were unique for each trajectory in the training and test data sets:

| Uniform Distribution | Parameters |
|:---:|:---:|
| [0,2] | $a_1, b_1, a_3, b_3$ |
| [-1,1] | $a_2, b_2, a_4, b_4, a_6, b_6, a_8, b_8$ |
| [-2,2] | $a_5, b_5, a_7, b_7$ |

This allowed for vector fields like the ones shown in the Figure 5 below.

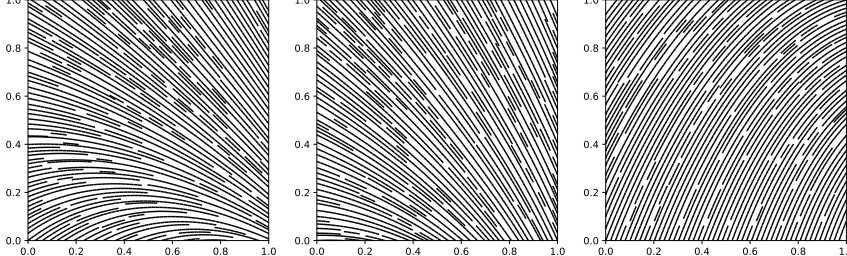

Figure 5: Three different and unique examples of the primary diffusion vector fields used for the Heat equation example in 3. Each example represents a diffusive field that was randomly determined for each trajectory

After all simulations were completed, a perturbed Delaunay triangulation grid was constructed and linear interpolation was used to generate the trajectories used for training at discretized points within the domain with a max cell measure of 0.0002 and from 0 to 20ms at every 1 ms intervals. In addition, all other data, like diffusion and the vector denoting the primary direction of diffusion orientation, were calculated at each discretized point for future uses. Training and test data sets consisted of 1000 and 100 trajectories, respectively.

### A.1.2 NETWORK DESCRIPTION FOR THE ANISOTROPIC HEAT EXAMPLE

For each trajectory, an undirected Laplacian matrix was created such that each point was connected to its 30 nearest neighbors and weighted using the heat kernel method described in the main text. The graph Laplacian was constructed using the NetworkX software and spectral decomposition was done using the Scipy software's `eigsh` function.

The network was trained and optimized using the PyTorch software with an $\ell_2$ loss. The network is described as so, 1. Linear mapping from 3 inputs (u, x, y) to 200 width. 2. 3 G-FuNK layers each with a width of 200 and using 50 modes and using a GELU activation function. 3. Two Linear layers going from a width of 200 to 32, with a GELU activation, and then 32 to 1. The network was integrated using the NeuralODE software with a forward Euler integrator with a time step of 0.25 ms. G-FuNK was optimized with an $\ell_2$ loss function and an Adam optimizer with an initial learning rate of $5 \cdot 10^{-4}$ with a step decayed learning rate which halved every 100 epochs. G-FuNK utilized early stopping and was stopped when the validation loss did not improve for over 50 epochs.

For the comparison models, FNO was trained via the code found at https://github.com/neuraloperator. The 2D FNO network is described as so 1. A linear layer lifting the solution, primary diffusion vector and positional coordinates from a width of 5 to 50, 2. next, 4 FNO layers using 10 modes for both directions with a width of 50 and all used a GeLU activation function and finally 3. a projection layer consisting of 2 linear layers with the first expanding to a width of 128 and the second to the output size of 1. For FNO, solutions were linearly interpolated onto a 2D unit grid discretized by 32 by 32 points. For the GNN, a message-passing neural network was used similar to Iakovlev et al. (2020) which consisted of 2 networks. For the weighted edge case, the network is described as so 1. A message network with an input width of 6 where solution at node $i$, solution difference between node $i$ and neighbors $j$, relative position between node $i$ and neighbors $j$ and difference in primary diffusion vectors at node $i$ and neighbors $j$ were used as inputs with a hyperbolic tangent activation function. There were no hidden layers used and next was an output linear layer of width 75. 2. An aggregation network consisting of an input linear layer with a width of 78 to include the message, the solution at node $i$, and the primary diffusion vector at node $i$, and a hyperbolic tangent function. This network has no hidden layers and next was an output linear layer which projected to a width of 1. The GNN used the 30 nearest neighbors as neighbors for each node and edges weights were calculated using the heat kernel described in 8. The GNN without edge weights is described as so 1. A message network with an input width of 6 where solution at node $i$, solution difference between node $i$ and neighbors $j$, relative position between node $i$ and neighbors $j$ and difference in primary diffusion vectors at node $i$ and neighbors $j$ were used as inputs. There were 3 hidden layers of width 75 with hyperbolic tangent activation functions and an output linear layer of width 75. 2. An aggregation network consisting of an input width of 78 to include the message, the solution at node $i$ and the primary diffusion vector at node $i$. This network has 3 hidden layers of width 75 with hyperbolic tangent activation functions and an output linear layer of width 1. Both the FNO and GNN networks were integrated using the NeuralODE software with a forward Euler integrator with a time step of 1 ms. Both used the same training protocol as G-FuNK.

### A.1.3 CONVERGENCE TEST

The model presented in the Heat Equation problem in Table 1 was trained and tested on different number of trajectories. As the number of trajectories increased the performance of the model increased as shown in Table 2 below.

| Training Sample, $M =$ | 50 | 100 | 250 | 500 | 750 |
|---|---|---|---|---|---|
| Rel. $\ell_2$ | .1001 | .0794 | .0641 | .0357 | .0292 |

Table 2: Numerical Convergence of G-FuNK for the Heat equation example.

Likewise, the model presented in the Heat Equation problem in Table 1 was trained and tested with different numbers of datapoints within the unit square domain with random anisotropic diffusion fields and the results are shown in Table 3 below.

| Datapoints per trajectory, $n_\alpha =$ | 256 | 1024 | 3952 |
|---|---|---|---|
| Rel. $\ell_2$ | .0502 | .0451 | .0357 |

Table 3: Numerical Convergence of G-FuNK for the Heat equation example.

## A.2  2D Reaction Diffusion

### A.2.1  Data Generation - Random Rectangle Example

For this section of work, the 2D Reaction Diffusion equations were solved using the openCARP software which used a finite element solver on a random rectangle to solve the Courtemanche equations. The solver used a Crank-Nicolson method with a time step of .2 ms and an average edge length of .2 mm. Each edge of the rectangle was drawn from an independent uniform distribution of [15,30] cm. The primary direction of diffusion was defined in the x-direction with a 5 to 1 anisotropic ratio and a base diffusion value of 0.0625 mS/mm.

After all simulations were completed, a down-sampled point cloud was calculated from the finite element mesh and nearest neighbor interpolation was used to generate the trajectories used for training at discretized points within the domain with a resolution of of one point per .25 mm voxel (via `open3d` voxel_down_sample_and_trace), and from 0 to 100ms at every 10 ms intervals. In addition, all other data like diffusion, vector orientation, etc. was calculated at each discretized point for future uses. Training and test data sets consisted of 1000 and 100 trajectories, respectively.

### A.2.2  Network Description for Random Rectangle Example

For each trajectory, an undirected graph Laplacian matrix was created such that each point was connected to its 30 nearest neighbors and weighted using the heat kernel method described in the main text. The graph Laplacian was constructed using the NetworkX software and spectral decomposition was done using the Scipy software's `eigsh` function.

The network was trained and optimized using the PyTorch software with a loss function with the form $||\hat{u} - u||_2 + 5 \cdot ||\nabla \hat{u} - \nabla u||_2$. This loss was used to minimize the error at the wavefront where the difference in the gradient is typically the largest. G-FuNK was optimized with the Adam optimizer with a step decay learning rate where the learning rate halved every 100 epochs starting at $5 \cdot 10^{-4}$. For this case, the eigenvalues used in $B$ in (10) were raised to powers 0, 1, and 2 for these cases. For both cases, $\theta_R$ was a full matrix of learnable parameters.

For the random rectangle example, the network is described as so, 1. Linear mapping from 1 input (u) to 200 width. 2. Three G-FuNK layers each with a width of 200 and using 50 modes and each using a GELU activation function. 3. Two Linear layers going from a width of 200 to 32, with a GELU activation, and then 32 to 1. The network was integrated using the NeuralODE software with a forward Euler integrator with a time step of 1 ms.

For the comparison models, Geo-FNO was trained via the code found on their repository: `https://github.com/neuraloperator`. The 2D Geo-FNO network is described as so 1. A linear layer lifting the solution, fiber vector and positional coordinates from a width of 5 to 50, 2. next, 4 FNO layers using 10 modes for both directions with a width of 50 and all used a GeLU activation function and finally 3. a projection layer consisting of 2 linear layers with the first expanding to a width of 128 and the second to the output size of 1. The latent space was defined as a unit square discretized in 32 by 32 points. For the GNN, a message-passing neural network was used similar to Iakovlev et al. (2020) which consisted of 2 networks as so 1. A message network with an input width of 6 where solution at node $i$, solution difference between node $i$ and neighbors $j$, relative position between node $i$ and neighbors $j$ and difference in primary diffusion vectors at node $i$ and neighbors $j$ were used as inputs. There were 3 hidden layers of width 40 with hyperbolic tangent activation functions and an output linear layer of width 40. 2. An aggregation network consisting of in input width of 43 to include the message, the solution at node $i$ and the primary diffusion vector at node $i$. This network has 3 hidden layers of width 40 with hyperbolic tangent activation functions

and an output linear layer of width 1. The GNN used the 30 nearest neighbors as neighbors for each node and edges weights were calculated using the heat kernel described in 8. Both the Geo-FNO and GNN networks were integrated using the NeuralODE software with a forward Euler integrator with a time step of 1 ms. Geo-FNO was trained using a $\ell_2$ loss and the GNN was trained using a relative $H_1$ loss function. Both used the same training protocol as G-FuNK.

### A.2.3 OUT-OF-DISTRIBUTION PREDICTIONS

The model used in Table 1 for the random rectangles example was tested on out-of-distribution geometries such that the y-axis edge length of the rectangle was set to 35 cm which resulted in 15 geometries with sizes ranging from $[15, 30]cm \times 35cm$ which were well outside of the training distribution which contained random edge lengths in the range of $[15, 30]cm \times [15, 30]cm$. While predicting on out-of-distribution geometries was not part of the problem set up, we demonstrate G-FuNK's performance when extrapolating on out-of-distribution geometries. The model preformed with a relative $\ell^2$ error of .2140. The results are shown in Figure 6 below.

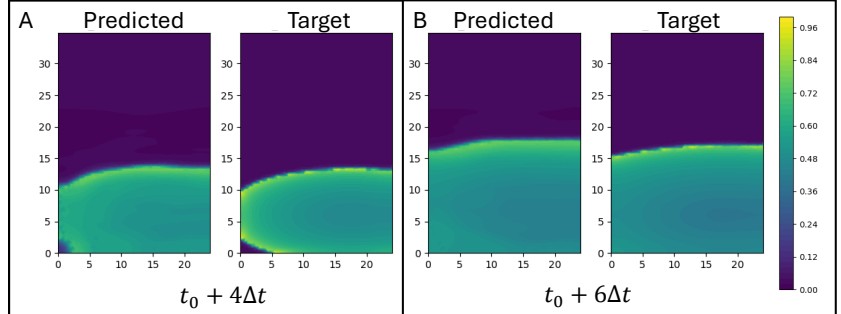

Figure 6: G-FuNK's performance on an out-of-distribution test set. A) At 40 ms after the initial condition. B) At 60 ms after the initial condition.

As expected, the error increases when predicting on out-of-distribution test geometries; this is primarily due to the difference in speed of the predicted and ground truth wave front. The shape of the propagating wave is predicted quite accurately as shown in the figure. Therefore, we anticipate that few-shot learning will be effective with a small fraction of the number of original training geometries included from the out-of-distribution domain to improve the pre-trained model's abilities on larger geometries.

### A.2.4 GEO-FNO PERFORMANCE ON ROTATED DATA

As described in the the Discussion, GEO-FNO was tested on $90°$ rotated data. Compared to the original data the netwrok was trained on, the performance drop to a relative $\ell^2$ error of 0.5681 when tested on the same data set but rotated. These results are shown in Figure 7 below.

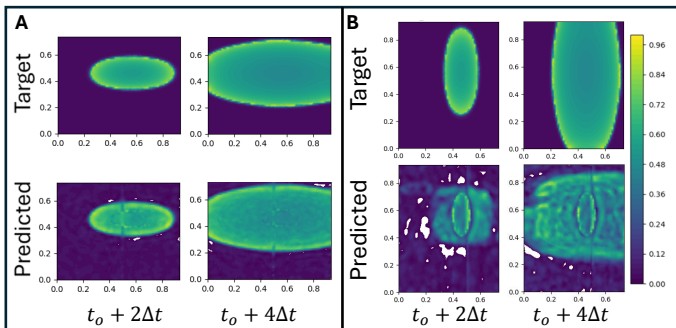

Figure 7: Geo-FNO Performance: Target vs Prediction for the random rectangle case. **A** Geo-FNO performance on the original test set. **B** Geo-FNO performance on the same original test set rotated 90 degrees. For **A** and **B**, the colorbar is min-max scaled from -85 to 20 mV and $\Delta t$ is 10 milliseconds.

## A.3 CARDIAC ELECTROPHYSIOLOGY

### A.3.1 DATA GENERATION FOR THE CARDIAC ELECTROPHYSIOLOGY EXAMPLES

From cardiac CT scans of 25 patients, the left atria were segmented and a surface mesh was extracted using the Seg3D software. Manual corrections were made using MeshMixer and average edge length was set to 0.25 mm. Surfaces were then annotated for calculating universal atrial coordinates for mapping atrial fibers with the atlas based method. The Courtemanche equations Courtemanche et al. (1998) were solved using the openCARP software which used a finite element solver which implemented a Crank-Nicolson scheme with a time step of 0.2 ms.

The diffusion tensor $\mathbf{K}(\mathbf{x})$ was calculated using the following equations.

$$\hat{\mathbf{F}}(\mathbf{x}) = \left[ \begin{array}{ccc} | & | & | \\ \hat{F}^{(1)}(\mathbf{x}) & \hat{F}^{(2)}(\mathbf{x}) & \hat{F}^{(3)}(\mathbf{x}) \\ | & | & | \end{array} \right] \tag{18}$$

$$\mathbf{K}(\mathbf{x}) = \hat{\mathbf{F}}(\mathbf{x}) \left[ \begin{array}{ccc} \sigma^{(1)} & 0 & 0 \\ 0 & \sigma^{(2)} & 0 \\ 0 & 0 & \sigma^{(3)} \end{array} \right] \hat{\mathbf{F}}^{T}(\mathbf{x}) \tag{19}$$

Where $\hat{F}^{(1)}$ is the mapped fiber field from the atlas-based method described in Roney et al. (2021); Ali et al. (2021); Roney et al. (2019), $\hat{F}^{(1)}$, $\hat{F}^{(2)}$, and $\hat{F}^{(3)}$ are orthonormal and $\sigma^{(2)}$ and $\sigma^{(3)}$ are the base diffusive coefficient of 0.625 mS/mm with $\sigma^{(1)}$ set as 5 times greater.

After all simulations were completed, a down-sampled point cloud was calculated from the finite element mesh and nearest neighbor interpolation was used to generate the trajectories used for training at discretized points within the domain with a resolution of one point per .4 mm voxel (via `open3d`'s function voxel_down_sample_and_trace), and the trajectory was downsampled from every 1 ms to every 10 ms in the range 0 to 300 ms. In addition, all other data like diffusion and the fibers vector orientation were calculated at each discretized point for future uses. For the multiple geometries case, training and test data sets consisted of 2400 and 100 trajectories, respectively where each geometry had 100 trajectories and the test set consisted of all 100 trajectories of a hold-out geometry.

### A.3.2 NETWORK DESCRIPTION FOR CARDIAC ELECTROPHYSIOLOGY EXAMPLES

For each trajectory, an undirected graph Laplacian matrix was created such that each point was connected to its 6 nearest neighbors and weighted using the heat kernel method described in the main text. The graph Laplacian was constructed using the NetworkX software and spectral decomposition was done using the Scipy software's `eigsh` function.

For this case, G-FuNK was trained and optimized using the PyTorch software with a relative $H_1$ loss function with the form $\frac{||\hat{u}-u||_2}{||u||_2} + \frac{||\nabla\hat{u}-\nabla u||_2}{||\nabla u||_2}$. This loss was used to minimize the error at the

wavefront where the difference in the gradient is typically the largest. An Adam optimizer equipped with a step decay learning rate with an initial rate of $5 \cdot 10^{-4}$ that halves every 100 epochs was used and was early stopped when the validation loss did not decrease after 50 epochs. The eigenvalues used in $B$ in (10) were raised to the powers $0, 1, 2$ in these cases. $\theta_R$ was a full matrix of learnable parameters.

For this case, the G-FuNK network is described as so, 1. Linear mapping from 1 input (u) to 300 width. 2. Three G-FuNK layers each with a width of 300 and using 25 modes and each using a GELU activation function. 3. Two Linear layers going from a width of 300 to 32, with a GELU activation, and then 32 to 1. The network was integrated using the NeuralODE software with a forward Euler integrator with a time step of 1 ms.

For the comparision GNN model, a message-passing neural network was used similar to Iakovlev et al. (2020) which consisted of 2 networks as so 1. A message network that operated on the edges with an input width of 8. The input features consisted of the solution at node $i$, solution difference between node $i$ and neighbor $j$, relative position between node $i$ and neighbor $j$, and difference in primary diffusion vectors at node $i$ and neighbors $j$ were used as inputs. There were 3 hidden layers of width 175 with hyperbolic tangent activation functions and an output linear layer of width 175. 2. An aggregation network consisting of an input width of 179 to include the message, the solution at node $i$ and the primary diffusion vector at node $i$. This network has 3 hidden layers of width 175 with hyperbolic tangent activation functions and an output linear layer of width 1. The GNN used the 6 nearest neighbors as neighbors for each node and edges weights were calculated using the heat kernel described in 8. The network was integrated using the NeuralODE software with a forward Euler integrator with a time step of 1 ms. A relative $H_1$ loss function was used to optimize the model and it implemented the same training protocol as G-FuNK.

### A.4 Other Examples

#### A.4.1 2D Reaction Diffusion with Random Fiber Fields and Anisotropic Diffusion Ratio

**Data Generation - Random Fiber Field and Ratio Example**    For this section of work, the 2D Reaction Diffusion equations were solved using the openCARP software which used a finite element solver on a 20 cm by 20 cm square to solve the Courtemanche equations. The solver used a Crank-Nicolson method with a time step of .2 ms and an average edge length of .2 mm. The primary direction of diffusion was defined as $F^1$ in the equation below.

$$F^{(1)}(x, y) = \begin{bmatrix} \frac{a_1 x}{L} + a_2 + \cos(\frac{a_3 \pi y}{L} + a_4) \\ \frac{b_1 x}{L} + b_2 + \sin(\frac{b_3 \pi y}{L} + b_4) \end{bmatrix} \tag{20}$$

The parameter $L$ was set to the edge length of the square, 20 cm. The parameters $a_1$ to $a_4$ and $b_1$ to $b_4$ were derived from the following uniform distributions and were unique for each trajectory in the training and test data sets:

| Uniform Distribution | Parameters |
|---|---|
| [.35,.65] | $a_1, b_1, a_2, b_2, a_3, b_3, a_4, b_4$ |

For this case, $\mathbf{K}$ is a diffusion tensor representing the anisotropy with the same form as (15) and (19) except the anisotropic ratio is a random parameter $\gamma$ drawn from a uniform distribution on $[4, 6]$ with a base diffusion of 0.0625 mS/mm. The diffusion was calculated using the equation 21 below.

$$\mathbf{K}(\mathbf{x}) = \begin{bmatrix} \hat{F}^{(1)}(\mathbf{x}) & \hat{F}^{(2)}(\mathbf{x}) \end{bmatrix} K_0 \begin{bmatrix} \gamma & 0 \\ 0 & 1 \end{bmatrix} \begin{bmatrix} \hat{F}^{(1)}(\mathbf{x}) & \hat{F}^{(2)}(\mathbf{x}) \end{bmatrix}^T, \tag{21}$$

Where $K_0$ is the base diffusive coefficient of 0.0625 mS/mm and $\gamma$ is a random parameter drawn from a uniform distribution of $[4, 6]$.

After all simulations were completed, a down-sampled point cloud was calculated from the finite element mesh and nearest neighbor interpolation was used to generate the trajectories used for training

at discretized points within the domain with a resolution of 250 and from 0 to 100ms at every 10 ms intervals. In addition, all other data like diffusion and vector orientation were calculated at each discretized point for future uses. Training and test data sets consisted of 1000 and 100 trajectories, respectively.

For the single geometry case, the network is described as so, 1. Linear mapping from 1 input (u) to 200 width. 2. Three G-FuNK layers each with a width of 200 and using 100 modes and each using a GELU activation function. 3. Two Linear layers going from a width of 200 to 32, with a GELU activation, and then 32 to 1. The network was integrated using the NeuralODE software with a forward Euler integrator with a time step of 1 ms.

The prediction vs. the ground truth for this example is shown in figure 8 below.

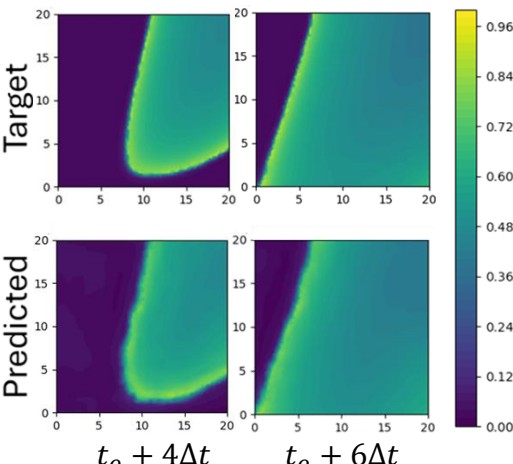

Figure 8: **Target (top)** vs **Prediction (bottom)** at different times after the initial condition $t_o$ for one domain in the random fiber and random anisotropic diffusion ratio case. The majority of the error is associated with the wavefront. $\Delta t$ is 10 milliseconds and the colorbar is min-max scaled from -85 to 20 mV.

For this case, the network used is described as so, 1. Linear mapping from 3 inputs (u, x, y) to 50 width. 2. Four G-FuNK layers each with a width of 50 and using 200 modes and each using a GELU activation function. 3. Two Linear layers going from a width of 50 to 32, with a GELU activation, and then 32 to 1. The network was integrated using the NeuralODE software with a forward Euler integrator with a time step of 1 ms.

### A.4.2 SINGLE ATRIAL EXAMPLE

G-Funk was trained on a single left atria using the same setup as the multiple EP example. The network was trained and tested on the same points and fiber fields with random initial conditions. The data consisted of training and test trajectories from time [0, 90] ms with 1000 and 100 trajectories, respectively.

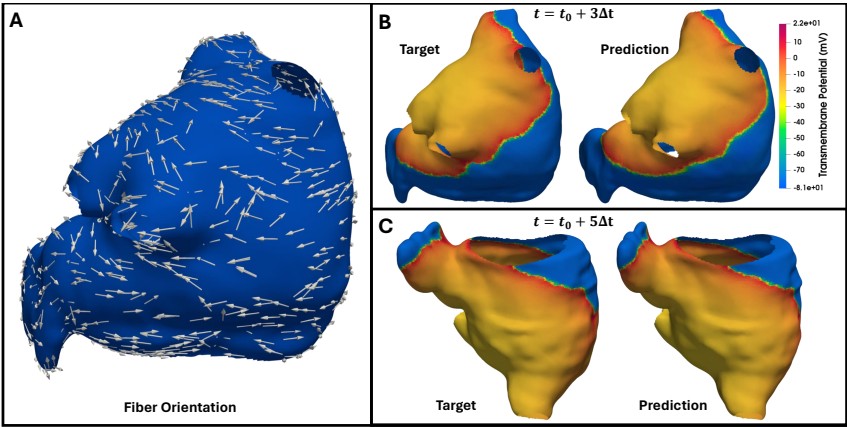

Figure 9: **A**: Vector representation of the fiber orientation across a left atrium. The direction of the white arrows is the primary direction of diffusion which influences the wave propagation. **B**: Posterior view of the LA at $3\Delta t$ (30 ms) after $t_0$ for the single geometry case. **C**: Anterior view of an LA at $5\Delta t$ (50 ms) after $t_0$ for the single geometry case.

For this case, the network is described as so, 1. Linear mapping from 1 input (u) to 100 width. 2. Three G-FuNK layers each with a width of 100 and using 25 modes and each using a GELU activation function. 3. Two Linear layers going from a width of 200 to 32, with a GELU activation, and then 32 to 1. The network was integrated using the NeuralODE software with a forward Euler integrator with a time step of 1 ms.

The network was trained and optimized using the PyTorch software with a relative $H_1$ loss function with the form $\frac{||\hat{u}-u||_2}{||u||_2} + \frac{||\nabla\hat{u}-\nabla u||_2}{||\nabla u||_2}$. This loss was used to minimize the error at the wavefront where the difference in the gradient is typically the largest. For this case, G-FuNK was optimized with a relative-$H_1$ loss function $\frac{||\hat{u}-u||_2}{||u||_2} + \frac{||\nabla\hat{u}-\nabla u||_2}{||\nabla u||_2}$. An Adam optimizer equipped with a step decay learning rate with an initial rate of $5 \cdot 10^{-4}$ that halves every 100 epochs was used and was early stopped when the validation loss did not decrease after 50 epochs. The eigenvalues used in $B$ in (10) were raised to the powers $0, 1, 2$ in these cases. $\theta_R$ was a full matrix of learnable parameters.

### A.4.3 MULTIPLE ATRIAL EXAMPLE

In the same set up as Cardiac Electrophysiology in 3, the model was allowed to predict from $t = [0, 190]ms$ which focused on the entire action potential cycle.

For the multiple geometries case from time [0, 190] ms, the network is described as so, 1. Linear mapping from 1 input (u) to 100 width. 2. Three G-FuNK layers each with a width of 100 and using 25 modes and each using a GELU activation function. 3. Two Linear layers going from a width of 200 to 32, with a GELU activation, and then 32 to 1. The network was integrated using the NeuralODE software with a forward Euler integrator with a time step of 1 ms.

The network was trained and optimized using the PyTorch software with a relative $H_1$ loss function with the form $\frac{||\hat{u}-u||_2}{||u||_2} + \frac{||\nabla\hat{u}-\nabla u||_2}{||\nabla u||_2}$. This loss was used to minimize the error at the wavefront where the difference in the gradient is typically the largest. For this case, G-FuNK was optimized with a relative-$H_1$ loss function $\frac{||\hat{u}-u||_2}{||u||_2} + \frac{||\nabla\hat{u}-\nabla u||_2}{||\nabla u||_2}$. An Adam optimizer equipped with a step decay learning rate with an initial rate of $5 \cdot 10^{-4}$ that halves every 100 epochs was used and was early stopped when the validation loss did not decrease after 50 epochs. The eigenvalues used in $B$ in (10) were raised to the powers $0, 1, 2$ in these cases. $\theta_R$ was a full matrix of learnable parameters.

### A.4.4 OTHER EXAMPLE RESULTS

| Problem | Geometries | Parameters | Training Trajectories | Metrics (Test Set) | Rel $\ell_2$ |
|---|---|---|---|---|---|
| 2D Reaction Diffusion | square | 12865 | $M$=1000 | Rel. $\ell_2$ | 0.0796 |
| Cardiac EP, $t = [0, 90]$ms | single | 327665 | $M$=1000 | Rel. $\ell_2$ | 0.0717 |
| Cardiac EP, $t = [0, 190]$ms | multiple | 35640 | $M$=2400 | Rel. $\ell_2$ | 0.0823 |

### A.5 COMPUTATIONAL RESOURCES

All data generation was completed using less than 12 CPU cores with less than 64 GB of RAM in no more than 25 minutes per trajectory (Computing time varies based on size of the mesh). Training of neural networks was completed using either a NVIDIA RTX A6000 with 48 GB of virtual ram or a NVIDIA RTX A4500 with 24 GB of virtual RAM. The Cardiac Electrophysiological example was the most extensive and took no more than 5 days to train.

