# OpenReview forum: "Graph Fourier Neural Kernels (G-FuNK): Learning Solutions of Nonlinear Diffusive Parametric PDEs on Multiple Domains"
_ICLR.cc/2025/Conference — Submitted to ICLR 2025_

### Official Review · Reviewer_KvZD · 2024-10-22

**Soundness:** 3
**Presentation:** 3
**Contribution:** 3
**Rating:** 8
**Confidence:** 3

**Summary:**

This paper proposes  Graph Fourier Neural Kernels (G-FuNK) for learning solution generators of time-dependent partial differential equations (PDEs) on graphs. G-FuNK aims to be "geometry-invariant" by leveraging the spectral domain of graphs through the Graph Fourier Transform (GFT), similar to how the Fourier Neural Operator (FNO) achieve "discretization-invariance" in regular domain.

**Strengths:**

The writing is clear, and the combination of graph neural networks (GNNs) and FNO is well motivated and novel to the best of my knowledge. The numerical examples show a gradual increase in complexity, leading to the final example on cardiac electrophysiology, which is complex and demonstrates the practical utility of the proposed framework.

**Weaknesses:**

In Table 1, for the three methods (G-FuNK, FNO, GNN), the number of parameters can differ by orders of magnitude, making it challenging to evaluate the improvement in performance.

The statement "Our method predicts entire trajectories in under 1 second, significantly outperforming traditional numerical methods. For example, cardiac EP simulations typically take at least 15 minutes on 12 CPU cores for one given set of initial conditions." could be more precise, particularly regarding the numerical solution being compared. It seems the cardiac EP simulations from numerical methods serve as the high-fidelity solutions that are used to generate the training data and evaluate the error. However, G-FuNK learns on reduced modes (k_max), which may lead to limited accuracy. It would be more informative if the numerical solutions are computed on coarser mesh that achieve similar of accuracy as G-FuNK, or using the reduced modes.

**Questions:**

Given that the paper focuses on PDEs on graphs, it seems that the "Multipole Graph Neural Operator for Parametric Partial Differential Equations" (MGKN), which claims to be mesh-invariant, would be a more suitable baseline for comparison. While the original MGKN framework does not explicitly tackle changing geometry, it seems it can still be applied in this setting, since only the graph is used as an input. There are also many similarities between these two approaches: the Fourier transform and inverse Fourier transform in G-FuNK play a role similar to kernel convolutions in MGKN, which are computed using the multipole algorithm. Additionally, both methods employ some form of truncation to make computation more tractable (e.g., limiting modes or long-range interactions). I would appreciate further discussion or comparison of these two approaches.

---

> ### Author Response · Authors · 2024-11-19
>
> Thank you so much for your support of our work. Below are our responses to weaknesses and questions.
>
> Weaknesses:
>
> In Table 1, for the three methods (G-FuNK, FNO, GNN), the number of parameters can differ by order of magnitude, making it challenging to evaluate the improvement in performance.
>
> Thank you for this observation, we would like to note that this was something we tried to work around but there were a few issues. First, for a FNO with a single Fourier layer, with a width of 50, there are already over 500k parameters. Additionally, for the GNN model, there was a bottle neck for the model to fit on the GPU while training and in addition we observed that making the GNN model larger only marginally improved its performance.
>
> The statement "Our method predicts entire trajectories in under 1 second, significantly outperforming traditional numerical methods. For example, cardiac EP simulations typically take at least 15 minutes on 12 CPU cores for one given set of initial conditions." could be more precise, particularly regarding the numerical solution being compared. It seems the cardiac EP simulations from numerical methods serve as the high-fidelity solutions that are used to generate the training data and evaluate the error. However, G-FuNK learns on reduced modes ($k_{max}$), which may lead to limited accuracy. It would be more informative if the numerical solutions are computed on coarser mesh that achieve similar of accuracy as G-FuNK, or using the reduced modes.
>
> This value should be more precise, the issue with giving a precise number is that the simulation time changes from patient to patient (i.e. bigger heart is a larger mesh and can take longer). A more precise answer is an average of 13.2 minutes ranging from 8.12 to 21.97 minutes. This information will be added to the manuscript.
>
> Questions:
>
> Given that the paper focuses on PDEs on graphs, it seems that the "Multipole Graph Neural Operator for Parametric Partial Differential Equations" (MGKN), which claims to be mesh-invariant, would be a more suitable baseline for comparison. While the original MGKN framework does not explicitly tackle changing geometry, it seems it can still be applied in this setting, since only the graph is used as an input. There are also many similarities between these two approaches: the Fourier transform and inverse Fourier transform in G-FuNK play a role similar to kernel convolutions in MGKN, which are computed using the multipole algorithm. Additionally, both methods employ some form of truncation to make computation more tractable (e.g., limiting modes or long-range interactions). I would appreciate further discussion or comparison of these two approaches.
>
> Thank you for this question. The MGKN relies upon message passing which uses an aggregation framework of the edges that was observed to inhibit a GNN to learn the effects of anisotropic diffusion on a traveling wave. We discuss this within the text on lines 132-134 and furthermore the GNN we presented as a comparison also uses a message passing framework, further presenting the need to develop a graph-based network that can better handle anisotropic data. The examples in that paper were only on a 1D problem and a 2D linear problem. We were not able to figure out a feasible way to adapt their code for 3D time-dependent dynamics. Additionally, with their method, it is difficult and sensitive problem to obtain block low rank multiscale matrices with nonideal partitions of the surface.

---

> > ### Comment · Reviewer_KvZD · 2024-11-21
> >
> > Thanks for the clarification.
> >
> > I understand the usual argument that "evaluating NO is very fast (about 1 second). The training time (about 1 day)  can be amortized in many-query settings". I think a more informative comparison could strengthen the argument. Specifically:
> > - Solve the PDE numerically with high accuracy (about 13min).
> > - (Option 1) Reduce the mesh size such that the numerical solution have the same accuracy as the NO, what is the runtime?
> > - (Option 2) Solve the linear system in the same reduced mode as the NO, what is the accuracy and time?
> >
> > In Figure 4, the traveling wave in the target solution has a much sharper transition region, and the peak of the solution is noticeably red (>10mV), but the prediction is has wider transition region and smaller magnitude (orange, <-10mV). Overall, the NO prediction seems more diffusive. What might be the reason? How to improve the lag of the wavefront (1.62 ms) ?
> >
> > I find this work interesting compared to many existing neural operator studies, as it tackles challenging 3D examples. Given the complexity of the problem, I see this as a proof of concept with potential for further development. That said, since I found myself draw to the cardiac EP example, but it is not my area of expertise, so I’ll lower my confidence level to 3.

---

> > > ### Author Response · Authors · 2024-12-02
> > >
> > > Thank you for your comments and interest in our work. To answer your question, we believe the majority of the errors in the cardiac example is due to the complexity of the example and the relatively small dataset (only 24 geometries). We do assume that as we increase the size of the training data that the error in the wavefront will decrease. We observed that with the addition of the H1 norm penalty in the loss function for this example we were able to better predict a sharper wavefront and will explore other penalties in the future. In the future we will explore option 1 that you suggested and we thank you for this suggestion.

---

### Official Review · Reviewer_1MCs · 2024-10-28

**Soundness:** 2
**Presentation:** 2
**Contribution:** 2
**Rating:** 5
**Confidence:** 3

**Summary:**

The paper proposed Graph Fourier Neural Kernels (G-FuNK), which aim to solve time-dependent, nonlinear partial differential equations (PDEs) with varying parameters and domains. G-FuNK leverages parameter- and domain-adapted spectral method. These operators are particularly well-suited for problems involving anisotropic diffusion and complex geometries. The paper demonstrates G-FuNK's effectiveness on several applications, such as heat equation simulations, reaction diffusion equations, and cardiac electrophysiology, showing promising results in terms of accuracy and computational speed.

**Strengths:**

1. G-FuNK leverages parameter- and domain-adapted spectral method such that it is well-suited for problems involving anisotropic diffusion and complex geometries.

2. The application on Cardiac Electrophysiology is very interesting.

3. The paper presents a detailed comparison with FNO and GNN methods, showing that G-FuNK can outperform these methods

**Weaknesses:**

1. The limitations should be addressed.

2. The so-called graph fourier transform is actually spectral method, which need the eigenvectors pre-calculated first. This procedure could make the method not useful for large scale problem. Computational efficiency and scalability should be reported, including offline computational for the eigenvectors and online computation time. Comparison with FNO and GNN in terms of efficiency is also absent.

3. The novelty of graph fourier transform is limited. This is a widely studyed area in graph neural network community.

**Questions:**

1. Why do the authors use the neural ODE  model? Any gain from the specific model?

---

> ### Author Response · Authors · 2024-11-19
>
> Thank you for reviewing our manuscript. We have address your comments below,
>
> Weaknesses:
> 1.	The limitations should be addressed.
> 2.	The so-called graph fourier transform is actually spectral method, which need the eigenvectors pre-calculated first. This procedure could make the method not useful for large scale problem. Computational efficiency and scalability should be reported, including offline computational for the eigenvectors and online computation time. Comparison with FNO and GNN in terms of efficiency is also absent.
>
> Thank you for this point, we agree that as the problem goes to a larger scale, the calculation of eigenvectors becomes more difficult. We report on line 212  that the computational complexity is of the order $O(k_{max}^2 n j)$ for computing the $k_{max}$ lowest eigenvectors of an undirected graph. Additionally, we are able to use a subsampled domain and predict the solution of a higher resolution domain which shows the advantage of using the global eigenvectors. The computation of the eigenvectors for our case takes on average 8.58 seconds  for the 200 lowest modes for each of the 25 cardiac atrial geometries (approximately 20k nodes in each graph). We report the online computation time for the predictions to be less than 1 second for the cardiac ep examples, in contrast to the numerical simulation time of anywhere from 8-22 mins per example. All three networks were similar in efficiency in that both training time and inference were similar and a discussion will be added to the paper to note this.
>
>
> 3.	The novelty of graph fourier transform is limited. This is a widely studyed area in graph neural network community.
>
> While we agree with the ubiquity of GFT, we propose that the novelty in our method comes from representing the PDE learning problem using the spectrum of the weighted Graph’s Laplacian for a more generalizable FNO that can also incorporate anisotropic fiber fields into the edges naturally (so that their effects on the solution field do not have to be directly learned by the network). Our comparisons demonstrate that for diffusive PDEs, using this approach instead of message passing GNN based methods is advantageous for including the fiber fields without increasing the number of input parameters in our network. In this regard, G-FuNK is very light weight.
>
> Questions:
>
> 1.	Why do the authors use the neural ODE model? Any gain from the specific model?
>
> Yes, we use the neural ODE framework for a few reasons. First, learning $\frac{du}{dt}$ via a neural ODE allows for easily applying Dirichlet boundary conditions since before integrating the points could simply be set to a value. Second, in cardiac electrophysiology (and many other systems) it is common to apply external stimulus at different timepoints so being able to arbitrarily apply a “non-learned” external stimulus is of a great advantage. Third, not being constrained to a constant time step is an advantage since at different time points within a simulation a smaller time step might be necessary for better stability. While using a neural ODE framework is not necessary in all cases, we believe it provides numerous advantages compared to the standard methods. We can also use different time stepping methods (implicit schemes for improved stability, higher order methods when necessary).

---

> > ### Comment · Reviewer_1MCs · 2024-11-20
> >
> > Thank you for the authors' responses. However, I still have some concerns:
> >
> > The authors encode the PDE parameters into the edge features of the graph Laplacian, which effectively corresponds to the finite difference discretization of  $-div(\kappa \nabla u)$  on regular grids. In this context, the so-called Graph Fourier Neural Kernels are essentially equivalent to a spectral method, so better performance is naturally expected. However, this raises an important question: what is the advantage of using this approach compared to a traditional spectral method, especially since the discretization and computation of eigenvectors are already in place?
> >
> > Efficiency is another significant concern. Could the authors provide more details to justify the computational efficiency of the proposed method?

---

> > > ### Author Response · Authors · 2024-11-21
> > >
> > > Thank you once again for these great questions.
> > >
> > > There are a few advantages to our approach over traditional spectral methods. Spectral methods suffer when there are very large gradients, sharp changes in parameters and very non-linear reaction terms which occur in the reaction-diffusion and cardiac EP examples. This is why there are currently only FEM solvers openly available for these equations. Since in the proposed method the G-FuNK layer uses a spectral and a local branch (top and bottom parts in the G-FuNK layer) the method is better suited to learn global and local interactions and effectively combines this information to learn the solution generator. Second, while in the examples we are learning solutions to certain PDEs, ultimately our method is data-driven so we can learn the dynamics of anisotropic diffusive systems from data alone without knowing any underlying equations. We have presented a novel neural operator which uses spectral methods that can learn the underlying physics of a system without the need of knowing an underlying system of equation necessary for traditional spectral methods.
> > >
> > > We would like to note two things, first, that the FEM software used is highly optimized for solving these sets of equations and second, that for the cardiac EP example the eigendecomposition took less than 9 seconds so G-FuNK with a total wall time of less than 10 seconds would still greatly outperform any traditional solver which takes anywhere from 8 to 22 minutes (depending on size of heart). In cardiac EP, it is common to perform parameter sweeps and test different set-ups (initial conditions) to determine an optimal treatment so being able to perform these parameter sweeps on completely new patients while only needing to perform eigendecomposition once is of great advantage in terms of computational efficacy and work overhead. While there is a computational overhead for initially training the model, the advantages to quickly infer accurate solutions on new unseen domains outweigh the computational demands of training G-FuNK.
> > >
> > > There is currently no spectral solving software available to solve the equations used in the reaction-diffusion and cardiac EP examples so unfortunately, we cannot directly compare the efficiency between our method and a spectral solver.

---

### Official Review · Reviewer_Hf3Q · 2024-10-29

**Soundness:** 2
**Presentation:** 2
**Contribution:** 2
**Rating:** 3
**Confidence:** 3

**Summary:**

This paper proposes a neural network model to learn a solution operator of time-dependent second-order semi-linear PDEs that takes diffusion tensor and random sample points of a family of domains as inputs.

**Strengths:**

The numerical demonstration is the cardiac EP example  is interesting since I understood the least. I will rely on an expert in this area to give a meaningful comment on this example.

**Weaknesses:**

This paper proposes a neural network model to learn a solution operator of time-dependent second-order semi-linear PDEs that takes diffusion tensor and random sample points of a family of domains as inputs.
While the first input has been considered by many authors, the difficulty here is to allow the domain to be chosen from a family of domains, denoted by $\\{\Omega_\alpha\\}_{\alpha\in \mathcal{A}}$. My first thought is whether the setup makes sense since there is no discussion on the class of domains that are imposed. I believe this won't work on arbitrary classes of domains. For example, e.g., for Riemannian manifolds, I would believe that if any pair of manifolds in this class have Riemannian metrics that are diffeomorphic (or even stronger, such as conformally equivalent), then the learning problem makes sense. One would need a notion of continuity between any pair of domains in the class, otherwise, it is not feasible to interpolate (to have a map that can interpolate between the training domains).  In the numerical examples shown in the paper,  there is an affine transformation between any pair of arbitrary side lengths in the 2D Nonlinear Reaction-Diffusion. For the Cardiac Electrophysiology, although the measured data come from 25 patients, the PDE is solved on processed domains (finally 24 of these), and I suspect that these domains are diffeomorphic.

The only interesting numerical demonstration is the cardiac EP example since I understood the least. I will rely on an expert in this area to give a meaningful comment on this example. In terms of methodology, I cannot understand why the Graph Laplacian structure is helpful, unless when the derivatives in Eq. (2) are defined with respect to the Riemannian metric of the embedded manifolds. It is also not obvious to me why the construction of the G-FUNK layer should be a way to go since I cannot reason it from any basic principle.

While some numerical results look interesting, I don't really understand why the approach should work in general due to the lack of theoretical justification. I am also not sure how the approach behaves if one increases the number of layers or parameters in each G-FUNK layer. Finally, the three numerical examples are low-dimensional problems (2D or 3D), I would naively believe the standard PDE solvers should be able to solve the problem accurately in a reasonable time. Solutions to these (FEM in the cardiac EP example) are being used to train the G-FUNK model.  Based on these concerns, I believe this paper is technically (or mathematically) not interesting and not suitable for publication in ICLR. I would urge the authors to consider submitting this work to a domain science journal that is relevant to personalized cardiac EP.

**Questions:**

NA

---

> ### Author Response · Authors · 2024-11-19
>
> Thank you for your review of our manuscript. We appreciate your input. Below we have responded to the comments in the review.
>
> -We will add to the manuscript the requirement that the domains $\Omega_{\alpha}$ must be diffeomorphic (we certainly do not need conformal equivalence and do not have conformal equivalence in any of the examples). Thank you for pointing out this missing detail. The domains do indeed need to be diffeomorphic. We do not provide a theoretical justification, based on the current state of the literature on operator learning theory, most results are for vanilla frameworks on fixed domains. There are rarely any theoretical guarantees for approximation error on model-specific variants of the canonical operator learning problem at least that we are aware of. In the simplest case, our method reduces to a standard Fourier Neural Operator for which there are universal approximation guarantees by Kovachki et al., but we can of course handle more general domains than just a regular square grid.
>
> -In the case of the heart surface, the PDE is being solved by finite elements on the surface and the anisotropic Laplacian term is defined on the surface, which is precisely what we aim to approximate with our graph Laplacian on the surface. The main point of G-FuNK is that the canonical Fourier construction is replaced by the graph Laplacian which adapts to both the diffusivity fields and the surface on which they are defined. The specific architecture is an analog to the Fourier Neural Operators.
>
> -The PDE solver for the 3D Cardiac EP example, which is an open-source fully optimized finite element library for these problems, takes roughly 13 minutes to execute for a single set of initial conditions. For applications of these models, one is interested in doing parameter sweeps across initial conditions and stimulus to find some optimal treatment procedures for patients. It is unclear to us what the reviewer means by these being low dimensional problems. To our knowledge, the literature on neural operators for PDEs seems to be primarily focused on 2D and 3D examples, which is what we provide benchmarks for.

---

> > ### Comment · Reviewer_Hf3Q · 2024-11-21
> >
> > Thank you for clarifying. As of point 3, I think solving PDE of low dimensional is usually has 1-3 dimensional spatial domain, otherwise it is high. One would consider NN when the PDE is high-dimensional since the classical method is subjected to curse of dimension. However, if one applies NN method to low-dimensional problems and compare to classical methods, the accuracy is usually poor (as shown in the Table 1 with error of 10%).
> >
> > While theoretical convergence is not available, can you numerically show some convergence for one of the examples, let's just pick the simplest example (2D heat equation). For example, check the numerical convergence rate as a function of training data and compare it with classical PDE solvers.
> >
> > Since you clarify that the key idea is replacing FFT with a transformation that maps to eigenvectors of Graph Laplacian as coordinates, if the domain is simple like 2D-box, one can indeed solve a Sturm-Liouville eigenvalue problem to attain analytic eigenfunctions (which is what you are trying to estimate with Graph Laplacians) such that you can replace FFT with an expansion over these eigenfunctions. I would expect that this will be the upper limit of the performance? Can you run a numerical experiment to confirm this?
> >
> > I believe these few numerical tests would clarify the advantage and limitations of the proposed schemes.

---

> > > ### Author Response · Authors · 2024-11-22
> > >
> > > Once again we thank the reviewer for their valuable input and have provided responses below.
> > >
> > > Q1) As of point 3, I think solving PDE of low dimensional is usually has 1-3 dimensional spatial domain, otherwise it is high. One would consider NN when the PDE is high-dimensional since the classical method is subjected to curse of dimension. However, if one applies NN method to low-dimensional problems and compare to classical methods, the accuracy is usually poor (as shown in the Table 1 with error of 10%).
> > > While theoretical convergence is not available, can you numerically show some convergence for one of the examples, let's just pick the simplest example (2D heat equation). For example, check the numerical convergence rate as a function of training data and compare it with classical PDE solvers.
> > >
> > > A)	Thank you for this suggestion, we have compared the numerical convergence for the heat example shown in example 1. It follows as so,
> > >
> > > Number of samples
> > >
> > > 50, 100, 250, 500, 750
> > >
> > > Rel. l2 error
> > >
> > > .1001, .0794, .0641, .0357, .0292
> > >
> > > So, it does seem that we have convergence as we increase the number of samples as expected. These results will be added to the manuscript. We do expect for example 2 and 3 that as we increase the amount of data that the error will decrease as we see in this numerical convergence test. For example 3 we presume that the 24 geometries we trained on may not be expressive enough for the distribution of left atrial geometries but generating these meshes and fiber fields (diffusion fields) are time consuming so we restricted ourselves to 24 (+1 test).
> > >
> > > Q2) Since you clarify that the key idea is replacing FFT with a transformation that maps to eigenvectors of Graph Laplacian as coordinates, if the domain is simple like 2D-box, one can indeed solve a Sturm-Liouville eigenvalue problem to attain analytic eigenfunctions (which is what you are trying to estimate with Graph Laplacians) such that you can replace FFT with an expansion over these eigenfunctions. I would expect that this will be the upper limit of the performance? Can you run a numerical experiment to confirm this?
> > >
> > > I believe these few numerical tests would clarify the advantage and limitations of the proposed schemes.
> > >
> > > A) The set of eigenfunctions is complete in L^2 (and therefore in subspaces of smoother functions, e.g. Sobolev spaces) and therefore its use does not limit the performance of our method. Of course, given the finite amount of data, only a finite dimensional space spanned by a finite number K of (lowest frequency) such eigenfunctions is used: this creates a bias in the model, which can be made arbitrarily small upon increasing K, but increasing K increases the variance in our estimators, so that a finite optimal K (which could be obtained upon cross validation) can be chosen. We can add this discussion to the manuscript.

---

> > > > ### Comment · Reviewer_Hf3Q · 2024-11-26
> > > >
> > > > Following up your answers. For part a) What about the rate for standard solver?
> > > >
> > > > As for part b), let me clarify my question using a simple geometry. Suppose your domain is a 2D-sphere embedded in R^3. Given uniform sampling data, we understand that the eigenfunctions of Laplace-Beltrami (which is what you are trying to estimate with the Graph-Laplacian framework) are simply spherical harmonics. Now, you are given a randomly sampled data lie on the sphere, and again, my understanding is that you are using the Graph-Laplacian to ultimately approximate spherical harmonics and subsequently employ G_Funk on data expressed in these coordinates. Intuitively, we expect the best performance is achieved when you are using the analytic spherical harmonics since your estimated spherical harmonics are subjected to errors (for spectral convergence error rate, see e.g. Belkin and Niyogi, or recent papers by Garcia-Trillos and Jeff Calder groups). What my question was: Can you verify numerically if you use the estimated eigenvectors, your results will converge to the result using analytic eigenfunctions, when both are using the same truncated eigenmodes.

---

> ### Author Response · Authors · 2024-12-02
>
> We would like to thank the reviewer for their comments and their time in reviewing our manuscript. Response to your questions are below.
>
> -For the heat equation example we used the software Mathematica which takes as an input the continuous domain, PDE and Initial/Boundary condition and solves the PDE to arbitrarily high precision. The rate of convergence is not applicable here since we do not pass in a discretization of the domain. (The numerical convergence examples were previously added to the Appendix)
>
> -There are several different errors which accumulate to give the final error (e.g., number of training samples, number of nodes in the graph, optimization error). The approximation of the eigenfunctions via the graph Laplacian is one of these errors and it is not likely that the analytical eigenfunction would improve the error more than the changes to the other factors mentioned above. In addition, our diffusion coefficient is not constant. Therefore, we don’t think it is feasible to compute the analytical eigenfunctions for each sample and since the purpose of this paper is to predict on anisotropic domains, we do not see the utility in these computations for our purposes.

---

### Official Review · Reviewer_uMgx · 2024-10-30

**Soundness:** 3
**Presentation:** 3
**Contribution:** 3
**Rating:** 3
**Confidence:** 3

**Summary:**

The authors proposes an interesting surrogate model that combines Graph as the discretization method for Fourier neural operators. I believe the manuscript may be accepted after a major revision.

**Strengths:**

The paper proposes merging graphs within the standard surrogate models, which allows estimating the solution of 2nd order PDEs with varying coefficients on complicated geometries.

**Weaknesses:**

I believe there are several important details missing in the manuscript. Below, I list them.

**Questions:**

**Major:**

- Is there an error estimator in the prediction? How can one trust the outcome of fitted G-FUNK model for an unseen problem?
- Is there a way to enforce conservation laws (or some notion of structure preservation) in prediction if the underlying PDE admits such constraint?
- Does the solution/data need to be smooth? Can you try out viscous burger's equation with emerging discontinuity as viscosity goes to zero?
- Is the outcome ODE stable? Is there a guarantee on stability of ODE?
- Can you compare the Data-Generation/Training/Prediction time of proposed method versus the ones from a standard finite element/volume solver in the presented test cases? I believe comparing only the time/complexity of prediction against standard solver is very much misleading.
- If we know the underlying PDE, wouldn't it make sense to incorporate that in the loss, similar to what PINN does?
- Can the author show a case of extrapolation? To me, similar to PINNs, the proposed method can only be used as an efficient interpolator within the space of training data. This makes me doubt the claims on "parameter and domain-adaptation". For example, if you train your model to estimate heat equation in 1d problem inside the domain [0,1], can you test it on domain [-10,10] with different boundary conditions?

**Minor:**

- Abstract: “… for which the highest-order term in the PDE is diffusive...” What does it mean? Do you mean the highest-order term is even or, it has to be 2? I’m guessing second-order PDEs, which needs to be clarified in Abstract.
- Abstract: "without the need for retraining" and not "without need for retraining”.

---

> ### Author Response · Authors · 2024-11-19
>
> We would like to thank the reviewer for their time and effort put into reviewing our manuscript. We addressed their comments.
>
> Major:
>
> W1) Some works provide theoretical approximation error for vanilla neural operator frameworks on a single domain, however, they do not provide generalization error bounds at all or explain the numerical performance to address. Even regarding theoretical results on approximation error, these generally are quite broad in terms of the class of models that permit and do not have specific rates. Model specific approximation error results are hard to come by, especially on problems defined on a family of domains.
>
> W2 and W6) Yes, incorporating a PDE loss, as in PINNs, is possible within this framework, but we chose not to include it to highlight the specific advantages of the G-FuNK layer. Adding a PDE loss would make it challenging to attribute the model's success to our method's novelties and significantly increase training time. For the cardiac EP example, implementing PINNs would be particularly complex due to the need for a local metric tensor or coordinate system on the 3D surface, as Euclidean space alone cannot represent the spatial derivatives. Additionally, parameterizing the multiply connected (N=5) topological domain of the cardiac geometry in $\mathbb{R}^3$ is non-trivial.
>
> Our approach is designed as a data-driven framework that can generalize across a family of PDEs without requiring explicit knowledge of the underlying equations. While our training data comes from simulations of known PDEs, the architecture leverages this indirectly, such as by incorporating diffusivity into the weighted graph. In the cardiac EP example, solution errors primarily arise from wavefront time delays, but the overall wavefront form remains physically consistent. PINNs are better suited for cases where physical principles cannot be learned from the data alone, but they may not be ideal for this application due to the challenges outlined above and their computational expense.
>
> W3) No, the solution does not need to be smooth. We present in both the reaction-diffusion and cardiac EP examples a solution where the “wavefront” is almost discontinuous and the PDE is nonlinear. Examples of this discontinuity are shown in Courtemanche et. al. It is expected that G-FuNK would perform similar to FNO/GEO-FNO in any examples such as the viscous Burgers equation previously shown in other works but with the advantage noted within this paper. We note that G-FuNK can be constructed to be identical to FNO but can also handle other cases.
>
> W4) This is a greatly discussed problem with neural ODEs and is an ongoing field of research. It should be noted that while the proposed work uses a neural ODE, G-FuNK can operate to predict the next time step like any other previous works have done. In particular, G-FuNK can be combined with an implicit solver, of course with significantly increased computational cost.
>
> W5) We appreciate the reviewer’s suggestion to compare data generation, training, and prediction times. The primary motivation behind G-FuNK is to accelerate clinical decision-making for new patient-specific geometries of the left atrium, where multiple pacing sites and frequencies must be tested to determine optimal ablation strategies.
>
> In a standard finite element solver workflow, each simulation for a patient-specific geometry takes approximately 13.5 minutes per pacing site. Testing multiple pacing sites for numerous patients quickly becomes computationally expensive. In contrast, G-FuNK requires a one-time offline training cost (1–2 days) on 24 patient geometries with varying pacing sites. Once trained, it can make predictions for new geometries and pacing configurations in seconds, enabling instantaneous parameter sweeps to determine optimal treatment strategies. This one-time effort amortizes over future test cases, significantly reducing the overall computational burden as the model generalizes without requiring retraining.
>
> We will clarify this in the revised manuscript and include a comparison of the computational costs in a clinical workflow, highlighting how G-FuNK achieves cost-effectiveness for large-scale simulation predictions, particularly for cardiac EP.
>
> W7) In example 2 and 3, we extrapolate to different domains and in example 1 and 3 we extrapolate to different parameters of the PDE. In fact, example 2 is almost exactly what you suggest with the different domains, but with varying sized 2D rectangles. The test domains for the reaction diffusion and cardiac EP examples are unseen in the training. The cardiac EP example shows a case of extrapolation and domain-adaptation as the test domain is on a never-before-seen geometry with completely different boundaries and domain. We also believe this example shows that it is parameter adaptable since the test geometry has its own unique diffusive field that is different from all the other domains used in training.
>
> Will fix minor issues.

---

> > ### Comment · Reviewer_uMgx · 2024-11-21
> >
> > I thank the authors for their responses. They clarified some of my doubts, but I still have a few remaining issues.
> >
> > > W2 and W6) ... Our approach is designed as a data-driven framework that can generalize across a family of PDEs without requiring explicit knowledge of the underlying equations....
> >
> > I see the motivation for data-driven estimator. However, in prediction we would like to guarantee some physical constraint, like mass, momentum, or energy conservation. Even having a PINN loss would not guarantee physical constraints in prediction. However, if you consider standard finite volume/element method, they guarantee some notion of conservation by design regardless of the underlying PDE. The proposed method does not seem to answer this doubt.
> >
> > >  In fact, example 2 is almost exactly what you suggest with the different domains, but with varying sized 2D rectangles....
> >
> > Can you please clarify the setting of example 2? What is the domain that is trained on, and what is the domain that is tested on? The text is a bit confusing to me. Are you training on a smaller domain and testing on a larger domain? Or you are considering the same domain size, by varying mesh spacing (which would make it an interpolation)?

---

> > > ### Author Response · Authors · 2024-11-22
> > >
> > > We thank this reviewer for the additional time they have spent reviewing our work. It is very appericated. Below we have address the previous comments.
> > >
> > > Q1) ... Our approach is designed as a data-driven framework that can generalize across a family of PDEs without requiring explicit knowledge of the underlying equations....
> > >
> > > I see the motivation for data-driven estimator. However, in prediction we would like to guarantee some physical constraint, like mass, momentum, or energy conservation. Even having a PINN loss would not guarantee physical constraints in prediction. However, if you consider standard finite volume/element method, they guarantee some notion of conservation by design regardless of the underlying PDE. The proposed method does not seem to answer this doubt.
> > >
> > > A1) We agree with your statement, and we currently do not implement any such conservations in our network. In current literature, most Neural Operators do not enforce these constraints but this is a new growing topic in the community. This is a future direction we would like to explore, and we thank you for your comment.
> > >
> > > Q2) In fact, example 2 is almost exactly what you suggest with the different domains, but with varying sized 2D rectangles....
> > > Can you please clarify the setting of example 2? What is the domain that is trained on, and what is the domain that is tested on? The text is a bit confusing to me. Are you training on a smaller domain and testing on a larger domain? Or you are considering the same domain size, by varying mesh spacing (which would make it an interpolation)?
> > >
> > > A2) In the second example we trained on multiple trajectories where each trajectory had a rectangular domain where the edge lengths were independently chosen in the range [15, 30]x[15,30] so each trajectory within the dataset has an associated rectangular domain that is randomly chosen (each $\Omega_\alpha$ is different). Likewise, in the test trajectories, each domain is rectangle and independently chosen in the same range [15, 30]x[15,30] but each trajectory has a unique domain and initial condition. So altogether, all trajectories in both the training and test sets have different domains which is why we claimed the method is domain adaptable.
> > >
> > > We apologize as we believe we misinterpreted the original question. We would like to note that accuracy in extrapolation is not guaranteed, which is in fact a rather typical phenomenon in many numerical techniques, including Neural Operators. We generated an out-of-distribution test trajectories for the random rectangle example where each edge of the rectangle were randomly picked from the ranges [15, 30]x35 (so that one length is in distribution and the other well outside the training distribution) and without retraining, the G-FuNK model used in example 2 was used to predict on these out-of-distribution trajectories and performance did decrease with a rel. l2 error of 21.4%. The majority of the error is associated with a slightly faster wavefront in the prediction compared to the target. While the performance did decrease in an out-of-distribution test case, we still believe that G-FuNK is parameter and domain-adaptable since it can generalize very well to domains and parameters that are in distribution. We can add this discussion to the manuscript but this kind of out-of-distribution domain adaptation was not a primary goal in this work.

---

> > > > ### Comment · Reviewer_uMgx · 2024-11-27
> > > >
> > > > I thank the authors for their response and rebuttal.
> > > >
> > > > >  We can add this discussion to the manuscript but this kind of out-of-distribution domain adaptation was not a primary goal in this work.
> > > >
> > > > I think this should be clarified in the paper.
> > > >
> > > > >  the G-FuNK model used in example 2 was used to predict on these out-of-distribution trajectories and performance did decrease with a rel. l2 error of 21.4%.
> > > >
> > > > Where should I look at in the manuscript? which test case is "example 2"? Where do you show that the rel. error is around 21%?

---

> > > > > ### Author Response · Authors · 2024-11-28
> > > > >
> > > > > We thank the reviewer for their time and effort in reviewing our manuscript. We agree that this should be added to the manuscript to clarify the limitations of G-FuNK. We have address this at the end of the discussion section.
> > > > >
> > > > > We have added to the manuscript and we specifically mention that G-Funk is only predicting on in-distribution samples and reference to the Appendix section where we added the example of the out-of-distribution test as the reviewer suggested.

---

> > > > > > ### Comment · Reviewer_uMgx · 2024-11-28
> > > > > >
> > > > > > I thank the authors for the revision. Just a few minor:
> > > > > >
> > > > > > p.16, line 821-828, reads " [15−30] ×35 cm " or "[15−30] ×[15−30] cm" Is the notation consistent with the rest of the paper? I'm concerned about the use of hyphen "-" for range. Second, if these are supposed to describe area, shouldn't the unit be also for area, like $cm^2$?
> > > > > >
> > > > > > p.16, line 851, should read "Therefore, we anticipate..." and not "Therefor, we anticipate...".
> > > > > >
> > > > > > p.16 line 851-855, "we anticipate that few-shot learning will be effective with a small fraction of the number of original training geometries included from the out-of-distribution domain to improve the pre-trained model’s abilities on larger geometries."
> > > > > >
> > > > > > I can see why "a small fraction of the number of original training geometries included from the out-of-distribution" may help with prediction. However, it would not be out-of-distribution test anymore. To me, it sounds like the network can only learn to interpolate within data set, and not really learning any underlying physics and the governing equation.

---

> > > > > > > ### Author Response · Authors · 2024-12-02
> > > > > > >
> > > > > > > Thank you to reviewer uMgx for their time and effort in reviewing our manuscript. All recommend minor changes should have been reflected in the manuscript.
> > > > > > >
> > > > > > > We would like to add one comment. Since the coefficient of the PDE in the spatial domain are varying (and the solution is space dependent) we cannot expect G-FuNK to extrapolate as well in these out-of-distribution examples. This is something not unique to G-FuNK but to most state of the art data-driven Neural Operators. The novelty of this work is a Neural Operator that can predict on varying coefficients and geometries at the same time something that has not been effectively addressed in the context of diffusive PDEs until now.

---

### Meta-Review · Area_Chair_FeAw · 2024-12-17

**Metareview:**

The paper introduces a novel family of neural operators, termed Graph Fourier Neural Kernels (G-FuNK), designed to learn the temporal dynamics of diffusive PDEs across multiple anisotropic domains with varying parameters. The method embeds geometric and directional information about the domains by combining graph Laplacian-based constructions for domain-specific components with non-adapted components learned from training data using FNO. Additionally, an integrated ODE solver is used to capture the system’s time evolution.

The paper presents an interesting approach and some compelling results, although a number of weaknesses were raised .First, there is a lack of theoretical justification for why the proposed method should work in general, particularly regarding the interplay between graph-based Laplacian constructions and the learned Fourier components. A clearer convergence analysis or performance comparison with standard methods (eg. FEM, etc..) would be necessary to substantiate some of the claims made in the paper. Second, reviewers noted that the paper lacks a quantitative understanding of the limitations of the Graph Laplacian FFT, which is central to the method. This raises concerns about scalability, stability, and general applicability beyond the presented examples. Finally, there is limited analysis regarding the impact of architectural choices (e.g., increasing the number of layers or parameters) on the method’s performance.

The panel ultimately recommends rejection. While the method demonstrates potential for domain-specific applications, the lack of theoretical justifications, convergence analysis, and scalability experiments limits its impact for the broader ICLR audience. That said, the contributions could be valuable in application-focused venues where solving PDEs on anisotropic domains is of high practical importance. The authors are encouraged to further develop the theoretical underpinnings of the approach, provide comprehensive comparisons to standard methods, and explore more comprehensively the scalability of the proposed method. With these improvements, this work could make a meaningful contribution to both the scientific computing and machine learning communities.

**Additional Comments On Reviewer Discussion:**

During the rebuttal period, the authors primarily focused on clarifying the motivations behind their method, elaborating on some of the design choices, and explaining how the proposed approach complements traditional methods such as finite element methods (FEM) and spectral methods. While these clarifications provided helpful insights into the intent and applicability of the work, they did not fully address the concerns regarding theoretical justification, convergence analysis, and quantitative comparisons to standard solvers.

---

### Decision · Program_Chairs · 2025-01-22

Reject